# Bimodal function of chromatin remodeler *Hmga1* in neural crest induction and Wnt-dependent emigration

Shashank Gandhi*, Erica J Hutchins, Krystyna Maruszko, Jong H Park, Matthew Thomson, Marianne E Bronner*

Division of Biology and Biological Engineering, California Institute of Technology, Pasadena, United States

**Abstract** During gastrulation, neural crest cells are specified at the neural plate border, as characterized by *Pax7* expression. Using single-cell RNA sequencing coupled with high-resolution *in situ* hybridization to identify novel transcriptional regulators, we show that chromatin remodeler *Hmga1* is highly expressed prior to specification and maintained in migrating chick neural crest cells. Temporally controlled CRISPR-Cas9-mediated knockouts uncovered two distinct functions of *Hmga1* in neural crest development. At the neural plate border, *Hmga1* regulates Pax7-dependent neural crest lineage specification. At premigratory stages, a second role manifests where *Hmga1* loss reduces cranial crest emigration from the dorsal neural tube independent of *Pax7*. Interestingly, this is rescued by stabilized ß-catenin, thus implicating *Hmga1* as a canonical Wnt activator. Together, our results show that *Hmga1* functions in a bimodal manner during neural crest development to regulate specification at the neural plate border, and subsequent emigration from the neural tube via canonical Wnt signaling.

*For correspondence:
shashank.gandhi@caltech.edu (SG);
mbronner@caltech.edu (MEB)

## Introduction

The neural crest is a vertebrate-specific stem cell population with the capacity to migrate long distances during embryonic development (*Bronner and LeDouarin, 2012*; *Le Douarin, 1980*; *Simões-Costa and Bronner, 2013*). Originating at the neural plate border, these cells occupy the leading edges of the closing neural folds during neurulation. Subsequently, premigratory neural crest cells that initially reside within the dorsal aspect of the developing neural tube undergo an epithelial-to-mesenchymal (EMT) transition in order to delaminate and migrate extensively. Upon reaching their terminal locations within the embryo, neural crest cells differentiate into a plethora of derivatives, including craniofacial cartilage, pigment cells, smooth muscle, and peripheral neurons and glia (reviewed in *Gandhi and Bronner, 2018*).

A feed-forward gene regulatory network (GRN) underlies the formation of the neural crest, from induction at the neural plate border to final differentiation into a multitude of cell types. This GRN is comprised of transcription factors and signaling pathways, partitioned into developmental modules (*Martik and Bronner, 2017*; *Simões-Costa et al., 2015*). Recently, new tools like single-cell RNA sequencing (scRNA-seq) and Assay for Transposase-Accessible Chromatin using sequencing (ATAC-seq) have enabled analysis of the neural crest GRN at a global level, helping to clarify lineage trajectories and elucidate key biological processes therein, ranging from proliferation to differentiation (*Williams et al., 2019*). These approaches have opened the way to extensive functional analysis of important nodes within the GRN, particularly at early stages of neural crest development, which are less well-studied.

Neural crest formation begins at the gastrula stage, with establishment of the presumptive neural ectoderm bordering the non-neural ectoderm. Quantitative gene expression analysis of gastrula

**eLife digest** The neural plate is a structure that serves as the basis for the brain and central nervous system during the development of animals with a backbone. In particular, the tissues at the border of the neural plate become the neural crest, a group of highly mobile cells that can specialize to form nerves and parts of the face. The exact molecular mechanisms that allow the crest to emerge are still unknown.

The protein Hmga1 alters how genes are packaged and organized inside cells, which in turn influences how genes are switched on and off. Here, Gandhi et al. studied how Hmga1 helps to shape the neural crest in developing chicken embryos. To do so, they harnessed a genetic tool called CRISPR-Cas9, and deleted the gene that encodes Hmga1 at specific developmental stages. This manipulation highlighted two periods where Hmga1 is active. First, Hmga1 helped to define neural crest cells at the neural plate border by activating a gene called *pax7*. Then, at a later stage, Hmga1 allowed these cells to move to other parts of the body by triggering the Wnt communication system.

Failure for the neural crest to develop properly causes birth defects and cancers such as melanoma and childhood neuroblastoma, highlighting the need to better understand how this structure is formed. In addition, a better grasp of the roles of Hmga1 in healthy development could help to appreciate how it participates in a range of adult cancers.

stage chick embryos has revealed a surprisingly high degree of overlap of multiple transcription factors associated with diverse cell types within single cells in the early neural plate border region, ranging from markers characteristic of the neural crest (*Pax7*), to neural (*Sox2*) and placodal (*Six1*) cell types (*Roellig et al., 2017*). This is consistent with the possibility that cells in the neural plate border are transcriptionally primed toward multiple cell fates, rather than committed to a particular lineage. What then leads to cell lineage commitment and specification toward neural crest rather than alternative fates, and to their subsequent ability to initiate migration from the neural tube? One possibility is that previously unidentified transcriptional and epigenetic regulators play a critical role in these processes.

In this study, we used scRNA-seq to identify novel transcription factors and chromatin remodelers expressed in neural crest cells of the early chick embryo. We first describe the single-cell transcriptome of early migrating neural crest cells emerging from the hindbrain, with a focus on identifying new transcriptional regulators. One of the genes uncovered in the neural crest cluster was *Hmga1*, a non-histone chromatin remodeler that has known roles in tumor metastasis (*Resar et al., 2018*), but has been understudied in development. We first characterized the expression and function of *Hmga1* during neural crest development using *in situ* Hybridization Chain Reaction (HCR) and observed *Hmga1* transcripts enriched in the neural crest, with the onset of expression preceding neural crest specification in the neural plate border. To test its functional role in neural crest development, we used plasmid- and protein-based CRISPR-Cas9 strategies to knock out *Hmga1* in neural crest progenitors with temporal precision. The results demonstrate an early role for *Hmga1* in neural crest lineage specification in a *Pax7*-dependent manner, resulting in the downregulation of neural crest specifier genes such as Snail2, *FoxD3,* and *Sox10*. Interestingly, loss of *Hmga1* after completion of neural crest specification revealed a distinct set of defects in cranial neural crest emigration and migration. Using in situ hybridization and a fluorescent protein-based reporter, we show that this is a consequence of reduced canonical Wnt activity mediated by *Wnt1*, which can be rescued by concomitantly expressing stabilized ß-catenin, thus establishing a separate role for *Hmga1* in delaminating neural crest cells as a Wnt pathway activator. Taken together, these results identify a dual role for *Hmga1* in neural crest development with an early effect on neural crest specification and a later effect on initiation of migration via the canonical Wnt signaling pathway, mechanisms that may be inappropriately redeployed during tumorigenesis.

## Results

### Single-cell RNA-seq of early migrating hindbrain neural crest reveals novel transcriptional regulators

Many RNA-seq datasets have sought to examine genes that are enriched in cranial neural crest cells compared with other tissues (*Simões-Costa et al., 2014*) or axial levels (*Martik et al., 2019*). However, here we aimed to identify highly expressed transcription factors and chromatin remodelers that may have been missed due to overlapping expression between neural crest cells and surrounding tissues. To this end, gastrula stage Hamilton Hamburger (HH) four embryos were electroporated with the neural crest enhancer FoxD3-NC2:eGFP and cultured ex ovo until stage HH12 (*Hamburger and Hamilton, 1951*). The NC2 enhancer labels early migrating neural crest cells (*Simões-Costa et al., 2012*), thereby facilitating dissection of the region surrounding the rhombomere (r) six migratory neural crest stream for dissociation (*Figure 1A–A′*). To aid downstream analysis and clustering, we introduced an 'outgroup' of dissected primary heart tube cells into the single-cell suspension and generated barcoded Gel Bead-In-Emulsions (GEMs) on the 10X Genomics platform. The library was sequenced at a depth of 50,000 median reads/cell to profile a total of 1268 cells, out of which 1241 cells passed the quality control filters (*Figure 1—figure supplement 1A–C*).

Following mapping and dimensional reduction, the cells split into distinct cellular subtypes (*Figure 1B*), including five cell types (mesoderm, otic, ectoderm, hindbrain, and neural crest) derived from the dissected tissue, and the spiked-in outgroup ('Heart tube'; $Myl2^+$, $Tnnt2^+$). Known genetic markers that were enriched in each population served to distinguish the neural crest subcluster ($Tfap2B^+$, $ItgB3^+$) from the surrounding tissues (i.e. otic placode ($Cldn3^+$, $Gbx2^+$); hindbrain ($Pax6^+$, $Zic2^+$); ectoderm ($Epcam^+$, $Crabp2^+$); mesoderm ($FoxC2^+$, $Col1A1^+$)) (*Figure 1—figure supplement 2A–B*). Consistent with the anatomical diversity of the mesoderm, the latter was further subdivided into specific cell types like myocardium ($Hand2^+$) and paraxial mesoderm ($Prrx1^+$) (*Figure 1—figure supplement 1D–E*). We particularly focused our subsequent analysis on the neural crest cluster in the context of the neighboring tissues of hindbrain and ectoderm (*Figure 1C,D,D′*).

We sought to determine all transcription factors and chromatin regulators that were expressed in the neural crest-specific subcluster, regardless of their expression in other cell types. To this end, we shortlisted all genes associated with the gene ontology terms 'DNA-binding', 'regulation of transcription', and 'transcription factor'. This revealed several chromatin remodelers and transcription factors with high levels of expression in neural crest cells (*Figure 1E*; *Figure 1—figure supplement 2C*). The identified genes fell into two groups, the first of which was comprised of transcription factors enriched in neural crest cells, with little overlap in surrounding cell types. As expected, many of these genes, including *Sox10*, *Ets1*, *MafB*, and *Nrip1*, are known for their expression in the neural crest (*Gandhi et al., 2020*; *Tani-Matsuhana et al., 2018*). Importantly, the second group was comprised of chromatin remodelers and/or transcriptional regulators previously overlooked in bulk transcriptomic datasets, including *Hmga1*, *Dact2*, *Ssrp1*, and *Tbxl1x*, due to overlapping expression in other tissues. Indeed, their distribution in low-dimensional space confirmed that a high proportion of cells in the hindbrain and/or ectoderm also expressed these genes (*Figure 1F*). The expression of a subset of the above genes was validated at HH12 (*Figure 1G–K*) by *in situ* hybridization chain reaction (HCR), which revealed an abundance of transcripts in both r4 and r6 neural crest streams that emerge from the hindbrain. Taken together, the results show that our single-cell gene expression analysis is sufficient to resolve the underlying heterogeneity of the chick hindbrain. We also identified several genes expressed in migrating neural crest cells not highlighted in previous datasets given their broad expression in other tissues.

### *Hmga1* is expressed in the neural plate, neural plate border, and neural crest cells

Of the novel transcriptional regulators that were previously overlooked in bulk transcriptomic datasets, we were particularly intrigued by the chromatin remodeler *Hmga1*, due to its extensively studied role in tumorigenesis. A member of the High Motility Group A (HMGA) superfamily, *Hmga1* encodes a small, nonhistone chromatin remodeling protein that binds to the minor groove of DNA, thereby affecting the chromatin landscape and facilitating the binding of other transcription factors in the opposing major groove (*Grosschedl et al., 1994*). In developing mouse embryos, Hmga1 has

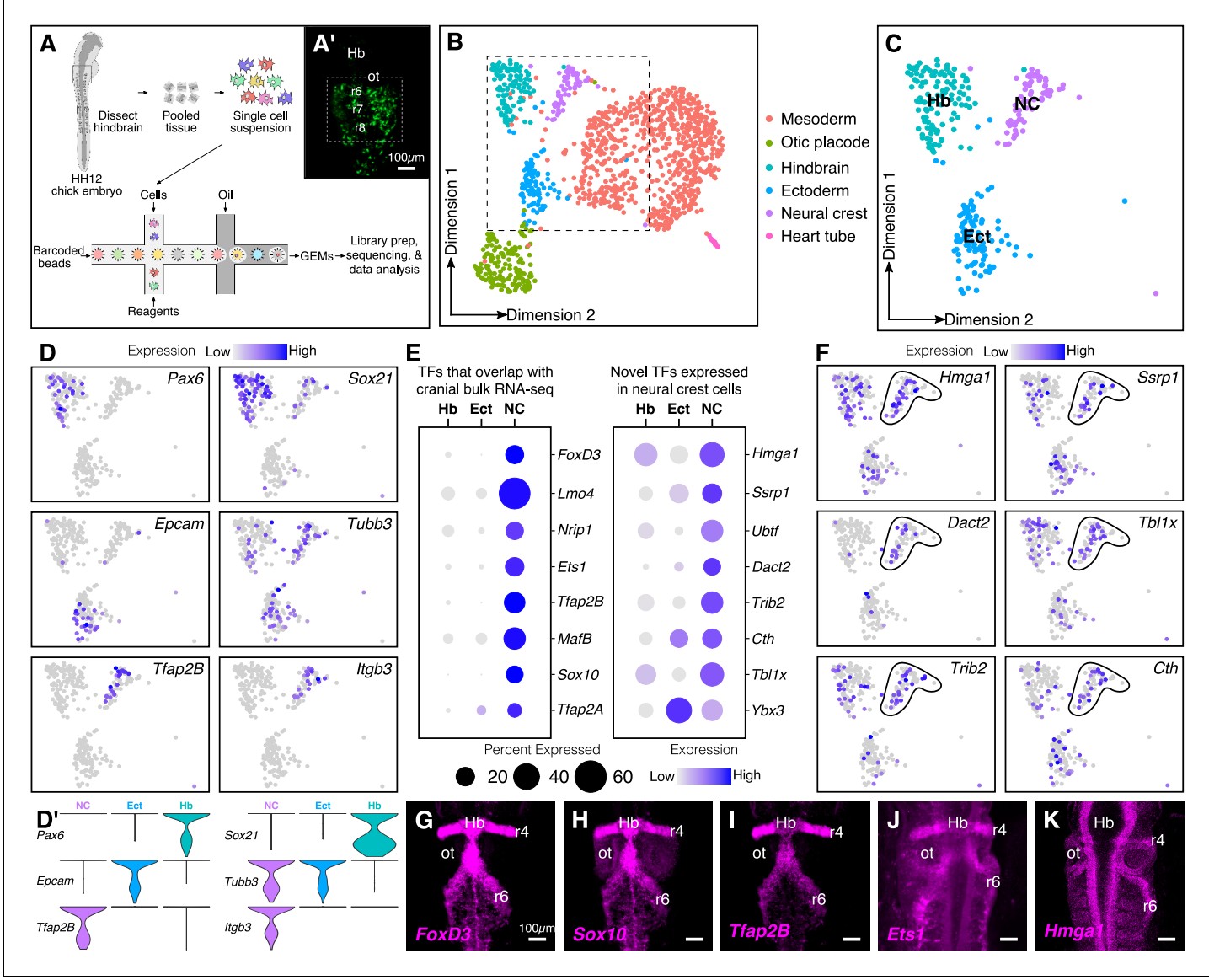

**Figure 1.** Single-cell (sc) RNA-seq of hindbrain neural crest reveals known and novel transcriptional regulators. (**A**) Schematic diagram illustrating the pipeline for performing scRNA-seq on the 10X Genomics platform. Reporter expression mediated by the FoxD3-NC2 enhancer (**A'**) was used as reference to dissect the hindbrain of HH12 chick embryos. Barcoded GEMs generated from the single-cell suspension were sequenced at a median depth of 50,000 reads/cell. (**B**) Dimensional reduction using UMAP identifies six subpopulations (including the spike-in) contained within the dissociated embryonic hindbrain. (**C**) Subset of B showing cells from hindbrain (Hb), ectoderm (Ect), and neural crest (NC). (**D–D'**) Feature plots used to visualize the expression of known marker genes as a means of identifying subpopulations in (**C**) in low-dimensional space. Single-cell expression distribution for marker genes (**D'**) in each cluster is shown as violin plots. (**E**) Genes that were associated with the GO terms 'DNA binding', 'regulation of transcription,' or 'transcription factor' were characterized as transcriptional regulators and the relative expression and abundance of a subset of them was visualized as a dot plot. The size of each dot corresponds to the percentage of cells expressing that specific gene in a given cluster, while the color represents the average expression level. (**F**) Feature plots showing expression of previously uncharacterized transcription factors or chromatin remodelers expressed in neural crest cells. (**G–K**) Hybridization chain reaction was used to validate the expression of a few factors that were identified in (**E**). Dorsal view of the hindbrain of HH12 shows migratory neural crest streams at r4 and r6 surrounding the otic. Hb, hindbrain; ot, otic placode; r, rhombomere; nc, neural crest; ect, ectoderm. See also *Figure 1—figure supplements 1* and *2*.

The online version of this article includes the following figure supplement(s) for figure 1:

**Figure supplement 1.** Quality of single-cell RNA-seq dataset.
**Figure supplement 2.** Identification of novel genes expressed in the neural crest.

been shown to have widespread expression across several tissues, including the brain, where its loss has been correlated with reduced developmental potential of neural precursor cells (*Kishi et al., 2012*). While well-studied in cancer (*Masciullo et al., 2003*; *Sarhadi et al., 2006*), little was known about its developmental function in neural crest cells. Given the parallels between the mechanisms that regulate delamination, migration, proliferation, and survival of neural crest cells and tumor cells (*Gallik et al., 2017*), we sought to characterize the expression of *Hmga1* during neural crest development.

At HH12, *Hmga1* was observed in migrating neural crest streams emanating from rhombomeres 4 and 6 (*Figure 1K*), and in the cranial mesenchyme, suggesting that its expression is not restricted to the hindbrain neural crest. Therefore, to determine its spatiotemporal pattern at early stages of neural crest development, we performed HCR for *Hmga1* at stages ranging from gastrulation (HH4), when neural crest cells are undergoing induction in the neural plate border (*Figure 2B'''*), to HH10, when neural crest cells have delaminated from the dorsal neural tube (*Figure 2E'''*) and are mid-migration in the cranial region (*Figure 2F'''*). As an early marker for the neural plate border and neural crest (*Basch et al., 2006*), we co-labeled *Pax7* transcripts at the aforementioned stages.

*Hmga1* transcripts were first detected in the neural plate and neural plate border, but not in the non-neural ectoderm at HH4+, and preceded the expression of *Pax7* in the neural plate border. *Hmga1* expression remained high at HH5-6 (*Figure 2A–B*), overlapping in the neural plate border (*Figure 2—figure supplement 1A*) with *Pax7* transcripts (*Figure 2B'*), as observed in transverse sections (*Figure 2B"*, *Figure 2—figure supplement 1B–B'*). As the neural plate border elevated to form neural folds between HH7 and HH8 (*Figure 2D'''*), expression of *Hmga1* was retained in the neural tube (*Figure 2C–D*) and continued to overlap with *Pax7* in the dorsal neural folds (*Figure 2D'*, *Figure 2—figure supplement 1C–D'*). Between stages HH9 and HH10, when neural crest cells delaminated from the dorsal neural tube (*Figure 2E'''*) and started migrating laterally (*Figure 2F'''*), *Hmga1* expression was retained in delaminating (*Figure 2E,E',E''*, *Figure 2—figure supplement 1E–F'*) and migrating (*Figure 2F,F',F''*, *Figure 2—figure supplement 1G–H'*) neural crest cells. Interestingly, transverse sections through a representative HH10 embryo revealed that within the migrating neural crest stream, *Hmga1* was expressed in both leader and follower cells, as compared to *Pax7*, which appeared to be downregulated in the leader cells (*Figure 2F''*). Together, these results show that the onset of *Hmga1* in the neural crest occurs in precursors at the neural plate border region prior to their specification and is retained in premigratory and migrating neural crest cells.

### *Hmga1* is necessary for neural crest specification

Given that *Hmga1* transcripts were enriched in the cranial neural crest, we sought to interrogate its possible role therein. To this end, we designed guide RNA plasmids (gRNAs) targeting the coding sequence of *Hmga1* (*Figure 3—figure supplement 1A*) and electroporated them together with constructs encoding Cas9 and nuclear RFP on the right side of HH4 gastrula stage embryos (*Figure 3A*). The left side of the embryo was electroporated with Cas9, nuclear RFP, and a control gRNA, chosen for its lack of binding in chick cells (*Gandhi et al., 2017*). Embryos were cultured ex ovo until stage HH9/9+ (*Figure 3B*), after which they were processed for immunohistochemistry, in situ hybridization, and HCR.

We first validated our knockout approach by probing for the expression of *Hmga1* itself in knockout embryos using HCR. This revealed a significant reduction in the abundance of *Hmga1* transcripts on the knockout side (*Figure 3C*). We quantified this phenotype in whole-mount embryos and observed a 25% reduction in *Hmga1* expression (*Figure 3D*, *Figure 3—source data 1*; p<0.05, Wilcoxon rank test). Notably, the loss of *Hmga1* transcripts in the neural crest was more dramatic than in the neural tube, due to targeted electroporation of knockout reagents to the presumptive neural plate border region. Next, we investigated the effect of knocking out *Hmga1* on *Pax7* expression in neural crest cells. We examined *Pax7* mRNA expression by HCR in embryos where *Hmga1* was knocked out on the right side, relative to the left side which served as an internal control. Consistent with their hierarchical onset of expression, loss of *Hmga1* resulted in a notable reduction in *Pax7* mRNA levels (*Figure 3E*). Next, we assessed whether the reduction in *Pax7* transcripts would consequently result in a loss of Pax7 protein in the neural crest by immunostaining *Hmga1*-knockout embryos with a Pax7 antibody. As expected, Pax7 protein levels were dramatically reduced in the migratory cranial neural crest (*Figure 3F,H'*), with further analysis revealing a significant decrease in

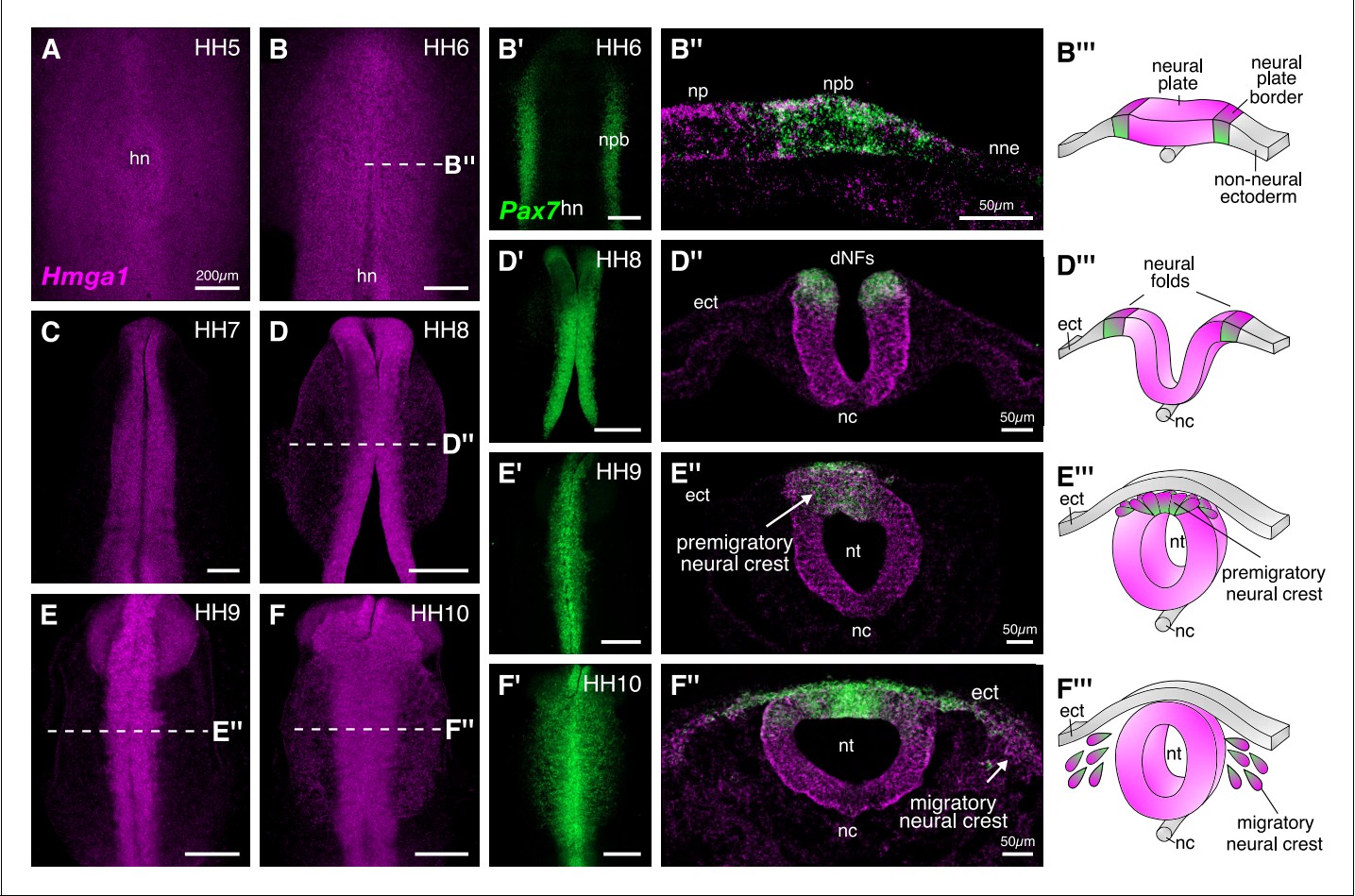

**Figure 2.** *Hmga1* is expressed in the neural plate, neural plate border, and premigratory and migratory neural crest cells. (A) HCR against *Hmga1* at HH5 reveals expression in the neural plate and neural plate border. (B–B') A wild-type HH6 chick embryo double-labeled with *Hmga1* (B) and *Pax7* (B') probes. *Hmga1* expression overlaps with *Pax7* in the neural plate border. (B''–B''') Transverse section through embryo in (B) shows *Hmga1* and *Pax7* transcripts in the neural plate and neural plate border, respectively, but not the non-neural ectoderm. (C–D') As the neural folds elevate, *Hmga1* expression is retained in the dorsal neural tube. (D''–D''') Transverse section through embryo in (D) shows *Hmga1* transcripts in the neural tube. (E–F) As neural crest cells delaminate (E''') and migrate laterally from the neural tube (F'''), *Hmga1* is expressed in emigrating (E'') and migrating neural crest cells (F''), along with *Pax7* (E' and F'). Arrow points towards delaminating (E'') and migrating (F'') cranial neural crest cells. hn, Hensen's node; npb, neural plate border; np, neural plate; nne, non-neural ectoderm; ect, ectoderm; dNF, dorsal neural folds; nc, notochord; nt, neural tube. See also *Figure 2—figure supplement 1*.

The online version of this article includes the following figure supplement(s) for figure 2:

**Figure supplement 1.** Overlapping expression of *Hmga1* and *Pax7* in the neural crest visualized in individual channels.

the number of Pax7+ cells on the knockout side compared to the control side (*Figure 3G*, *Figure 3— source data 2*; p<0.0001, student's t-test). Moreover, in the absence of *Hmga1*, cranial neural crest cells failed to migrate properly, as depicted by the expression of the migratory neural crest marker HNK1 (*Figure 3—figure supplement 1B*). A transverse section through the hindbrain (*Figure 3H*) of another representative *Hmga1*-knockout embryo stained for Pax7 (*Figure 3H'*) revealed a notable reduction in the expression of Pax7 (*Figure 3I*) as well as the neural crest specifier Snail2 on the knockout side (*Figure 3I'*). Furthermore, using in situ hybridization, we found that other neural crest specifier genes including *FoxD3* (*Figure 3J,J'*, *Figure 3—figure supplement 1D,E',E''*), *Tfap2b* (*Figure 3K*), *Sox10* (*Figure 3L,L'*), and *c-Myc* (*Figure 3—figure supplement 1C*) were also downregulated on the knockout side. In transverse sections through *Hmga1*-knockout embryos labeled for *Sox10* expression (*Figure 3L*), we also detected fewer Pax7+ cells (*Figure 3N*) and diminished levels of HNK1 expression (*Figure 3M*). On the other hand, no notable difference in the thickness of the

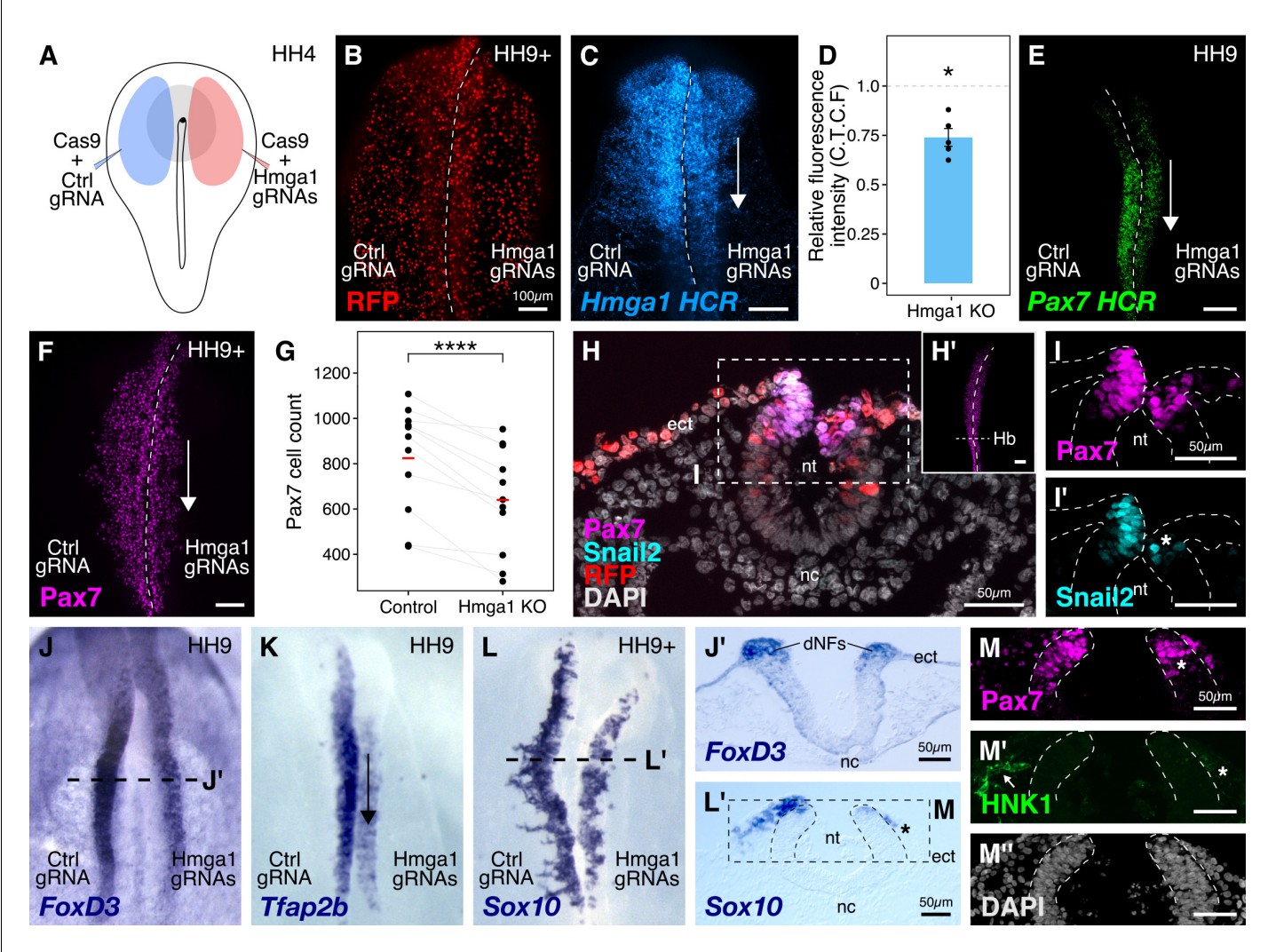

**Figure 3.** *Hmga1* knockout results in loss of neural crest specification. (**A**) The electroporation strategy for knocking out *Hmga1* using CRISPR-Cas9 in gastrula stage chick embryos. (**B**) Electroporated embryos were allowed to develop until HH9+ and screened for the expression of H2B-RFP. (**C**) Electroporation of Cas9 and *Hmga1* gRNAs on the right side resulted in loss of *Hmga1* transcripts in the neural crest as confirmed using HCR. (**D**) *Hmga1* expression in the neural crest quantified as corrected total cell fluorescence (CTCF) intensity in wholemount *Hmga1*-mutant embryos processed for HCR. A significant reduction in expression was observed (p-value<0.05, Wilcoxon rank test) on the experimental compared to the control side. A ratio of 1 (dotted line) corresponds to similar levels of *Hmga1* expression on both sides. (**E–F**) *Hmga1* knockout results in reduced *Pax7* expression in the neural crest, likely resulting from a significant reduction in *Pax7*+ cell count (**F**) on the knockout compared to the control side (****p<0.0001, student's t-test). (**G–H**) Transverse section through the hindbrain of a representative knockout embryo (**G'**) was stained for Pax7 (**H**) and the neural crest specifier Snail2 (**H'**). (**I**) *Hmga1* knockout also resulted in a reduction of *Pax7* transcripts on the knockout side. (**J–L**) *Hmga1*-mutant embryos were processed for in situ hybridization against neural crest specifier genes *FoxD3* (**J**, **J'**), *Tfap2b* (**K**), and *Sox10* (**L**, **L'**). (**M**) Transverse section through a representative embryo probed for the expression of *Sox10* showed reduced expression of the migratory neural crest marker, HNK1. The expression of Pax7 (**M'**) was also reduced, while the thickness of the neural tube remained unchanged (**M''**). See also *Figure 3—figure supplement 1*; *Figure 3—source data 1*, *Figure 3—source data 2*.

The online version of this article includes the following source data and figure supplement(s) for figure 3:

**Source data 1.** *Hmga1* HCR intensity whole mount embryos compared between Control and *Hmga1*-knockout sides in panel C.
**Source data 2.** Pax7-positive cell counts in whole mount embryos compared between Control and *Hmga1*-knockout sides in panel G.
**Figure supplement 1.** Knocking out *Hmga1* in gastrula stage embryos.

neural tube was observed (*Figure 3H,J',M''*, *Figure 3—figure supplement 1E*), suggesting that the targeted knockout of *Hmga1* in the neural plate border affected the neural folds/dorsal neural tube but not the neural plate itself. Taken together, our results indicate that *Hmga1* is important for proper specification of neural crest cells.

## Hmga1 regulates expression of *Pax7* in neural crest precursors in the neural plate border

As the expression of *Hmga1* precedes that of *Pax7* in the neural plate border, and its loss also causes a reduction of *Pax7* levels in neural crest cells (*Figure 3F*), we next asked if this regulation occurs in the neural crest precursors that reside in the neural plate border. In the preceding experiments, we characterized the expression of neural crest markers following CRISPR-plasmid-mediated loss of *Hmga1* at stages corresponding to early stages of neural crest emigration and migration. However, one caveat to our plasmid knockout strategy is that it takes time after electroporation of the CRISPR constructs in gastrula stage embryos for Cas9 to transcribe, translate, and properly fold to form a functional Cas9-gRNA complex. As specification of neural crest cells at the neural plate border is ongoing at HH4, this means that functional Cas9 would not be available until well after initial electroporation. Given that neural crest development occurs in a rostral-to-caudal wave along the body axis (*Gandhi and Bronner, 2018*), we speculated that effects of plasmid electroporation would be more penetrant in the hindbrain compared to anterior regions of the embryo, such as the midbrain. To test this possibility, we generated transverse sections through the midbrain (*Figure 4A*) and hindbrain (*Figure 4B*) of *Hmga1*-knockout embryos and quantified the number of Pax7+ cells as a ratio of cell count on the experimental versus control sides. Accordingly, we noticed that while this ratio was $0.948 \pm 0.036$ (n = 5) at the midbrain level, it was significantly reduced in the dorsal hindbrain, with a mean ratio of $0.69 \pm 0.08$ (n = 5; $p<0.05$, paired Student's t-test) (*Figure 4C*, *Figure 4—source data 1*). These results supported our assumption that specification may have already occurred at the midbrain level by the time Cas9 was functionally active.

To circumvent this issue and test the earliest effects of knocking out *Hmga1* concomitant with its onset of expression, we turned to an alternative CRISPR knockout strategy that enabled the loss of *Hmga1* immediately after transfection. To this end, we electroporated recombinant Cas9 protein with in vitro-transcribed *Hmga1* or control gRNA as ribonucleoprotein (RNP) complexes on the right and left sides of gastrula stage embryos, respectively (*Figure 4D*), and cultured embryos ex ovo until HH6. First, to validate the Cas9-protein-mediated knockout strategy, we labeled *Hmga1* transcripts in knockout embryos using HCR and observed a very efficient reduction in *Hmga1* expression (*Figure 4E*), especially in the neural plate border, thereby offering precise temporal control over the loss of this gene's activity. Next, we assayed for the expression of Pax7 in the neural plate border by immunostaining *Hmga1*-knockout embryos and found that the levels of Pax7 protein in the neural plate border (*Figure 4F*, *Figure 4—figure supplement 1A*) were severely downregulated. Transverse sections through the experimental compared with control sides revealed that neural plate border cells no longer expressed Pax7 after the loss of *Hmga1* (*Figure 4H,I*), and that this was not a result of premature apoptosis or aberrant cell proliferation (*Figure 4—figure supplement 1B–G'*, *Figure 4—figure supplement 1—source data 1*). In further support of the latter, the thickness of the neural plate border remained unchanged (*Figure 4H',I'*). We also quantified the corrected Pax7 total fluorescence intensity (C.T.C.F) in the neural plate border and observed a statistically significant difference between the control and knockout sides ($p<0.01$, paired student's t-test), with mean Pax7 intensity on the control side being $89.853 \pm 23.388$ a.u. (n = 5) as compared to $42.763 \pm 16.079$ a.u. (n = 5) on the knockout side (*Figure 4J*, *Figure 4—figure supplement 1H*, *Figure 4—source data 2*). Taken together, these results suggest that *Hmga1* is required for the expression of Pax7 in neural crest precursors that are induced in the neural plate border.

## Hmga1 is not required for expression of neural plate border genes *Msx1* or *Tfap2a*

The neural plate border was initially thought to contain discrete domains corresponding to neural, neural crest, placodal, and epidermal precursors. However, recent work has demonstrated that cells in this region co-express genes characteristic of different cell fates and exhibit a broad developmental potential, suggesting they are not restricted to individual cell fates until later in development

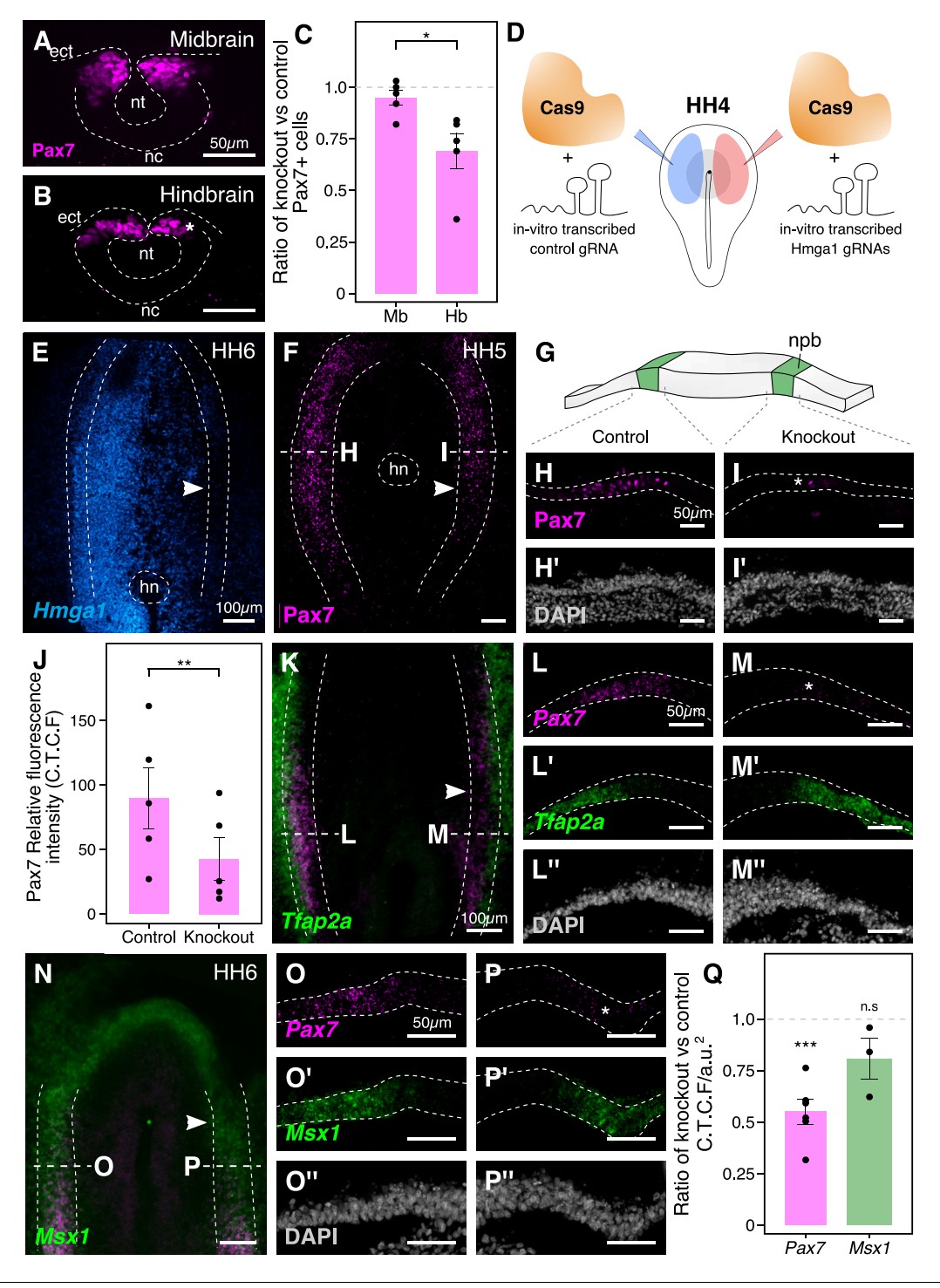

**Figure 4.** The effects of *Hmga1* knockout on neural crest specification are Pax7-dependent. (**A-B**) Transverse sections through a representative embryo show a dramatic reduction in the number of Pax7+ cells in the hindbrain (**B**, asterisk) as compared to the midbrain (**A**) at HH9/9+. As the hindbrain develops later than the midbrain due to the anterior-posterior progression of neural development, the effect on neural crest specification is more penetrant in the hindbrain (asterisk) due to the time lag between Cas9 plasmid electroporation and its activation in transfected cells. (**C**) The ratio of Pax7+ cells between the experimental and control sides quantified at the midbrain and hindbrain levels is significantly different (*p<0.05; student's t-test). A ratio of 1 (dotted line) corresponds to a similar number of Pax7+ cells on both sides. (**D**) Electroporation strategy for knocking out

*Figure 4 continued on next page*

*Figure 4 continued*

*Hmga1* using Cas9 protein and in vitro-transcribed gRNAs. This strategy was used to immediately reduce the levels of *Hmga1* on the knockout side. (**E**) HCR against *Hmga1* in mutant embryos shows dramatic transcriptional reduction on the experimental side (arrowhead). (**F**) Cas9-protein-mediated loss of *Hmga1* resulted in downregulation of Pax7 expression in the neural plate border on the right side (experimental side; arrowhead). (**G**) Illustration of the neural plate border. (**H–I'**) Transverse section through embryo shown in **F**. Electroporation of the control ribonucleoprotein (RNP) complex had no effect on the expression of Pax7 in the neural plate border (**H**), whereas the knockout side showed an almost complete loss (**I**, asterisk). No difference in the thickness of the neural plate border was observed between the two sides (**H',I'**). (**J**) Quantification of relative fluorescence intensity for Pax7 signal calculated as corrected total cell fluorescence (C.T.C.F) revealed a statistically significant difference between the control (left) and knockout (right) sides (**p<0.01, paired student's t-test). (**K–P''**) Representative *Hmga1*-mutant embryos that were processed for HCR against neural plate border genes *Tfap2a* (**K**; experimental side - arrowhead) and *Msx1* (**N**; experimental side - arrowhead). While *Hmga1* loss resulted in reduction of *Pax7* transcripts on the experimental sides (**M,P**; asterisk) compared to the control sides (**L,O**), the expression of *Tfap2a* (**L',M'**) and *Msx1* (**O',P'**) was relatively unchanged. No notable difference was observed in the thickness of the neural plate border (**L'',M'',O'',P''**). (**Q**) Transcriptional response to the loss of *Hmga1* was quantified as the ratio of knockout versus control C.T.C.F per unit area. While *Pax7* expression was significantly reduced (***p<0.001, paired student's t-test), no significant difference in *Msx1* expression was observed (n.s. p>0.05, paired student's t-test). Dotted line represents unperturbed ratio. See also *Figure 4—figure supplement 1*; *Figure 4—source data 1*, *Figure 4—source data 2*, *Figure 4—source data 3*.

The online version of this article includes the following source data and figure supplement(s) for figure 4:

**Source data 1.** Pax7-positive cell counts in transverse sections through the midbrain and hindbrain compared between Control and *Hmga1*-knockout sides in panel C.

**Source data 2.** Pax7 corrected total cell fluorescence intensity in transverse sections through the neural plate border compared between Control and *Hmga1*-RNP-knockout sides in panel J.

**Source data 3.** Pax7 and Msx1 HCR corrected total cell fluorescence intensity in transverse sections through the neural plate border compared between Control and *Hmga1*-RNP-knockout sides in panel Q.

**Figure supplement 1.** *Hmga1* knockout does not affect neural crest cell proliferation and apoptosis.

**Figure supplement 1—source data 1.** Pax7, Cleaved-caspase-3, and phospho-histone-H3 corrected total cell fluorescence intensity in transverse sections through the neural plate border compared between Control and *Hmga1*-RNP-knockout sides in panel H.

(*Roellig et al., 2017*). In chick embryos, *Tfap2a* and *Msx1* are expressed in the neural plate border, with *Tfap2a* transcripts spanning both the neural plate border and the non-neural ectoderm (*de Crozé et al., 2011*; *Luo et al., 2003*), whereas *Msx1* transcripts are expressed within a subset of Pax7+ cells in the neural plate border region (*Khudyakov and Bronner-Fraser, 2009*). Given that loss of *Hmga1* resulted in a reduction in Pax7 protein levels in the neural plate border, we asked if this was a result of a general neural plate border defect versus a selective effect on *Pax7*. To test this, we examined the expression of *Tfap2a* (*Figure 4K*) and *Msx1* (*Figure 4N*) transcripts together with *Pax7* using HCR in *Hmga1*-knockout embryos developed to neurula stages. Consistent with the loss of Pax7 protein in the neural plate border, *Hmga1* knockout caused reduced *Pax7* mRNA levels on the experimental (*Figure 4M,P*) compared to the control (*Figure 4L,O*) sides. However, the expression of *Tfap2a* (*Figure 4L',M'*) and *Msx1* (*Figure 4O',P'*) was retained in the absence of *Hmga1*, together with no noticeable difference in the thickness of the neural plate border (*Figure 4L'',M'',O'',P''*). Since *Pax7* and *Msx1* are specifically expressed in the neural plate border, we quantified the corrected total cell fluorescence intensity per unit area associated with their transcripts and calculated the ratio between the experimental and control sides. Under control conditions, this ratio would be close to 1. However, for *Pax7*, the mean calculated ratio was $0.552 \pm 0.06$ (n = 6), with a statistically significant difference between the experimental and control sides (p<0.001, paired Student's t-test). On the other hand, while the mean calculated ratio for *Msx1* was $0.809 \pm 0.098$ (n = 3), the fluorescence intensities were not significantly different between the two sides (*Figure 4Q*, *Figure 4—source data 3*). Taken together, these results show that *Hmga1* specifically regulates *Pax7* expression at the neural plate border.

## Hmga1 and Pax7 rescue the effects of losing *Hmga1* on neural crest specification

Given that loss of *Hmga1* affects neural crest specification, we asked if its overexpression would have the converse effect. To exogenously provide Hmga1, we designed a plasmid construct containing the coding sequence of *Hmga1* under the regulation of a ubiquitous enhancer/promoter combination (*Figure 5A*). This construct also contained the coding sequence for nuclear RFP downstream of an Internal Ribosome Entry Site (IRES), allowing identification of successfully transfected cells. To test the effect of overexpressing Hmga1, we electroporated this construct on the right side of

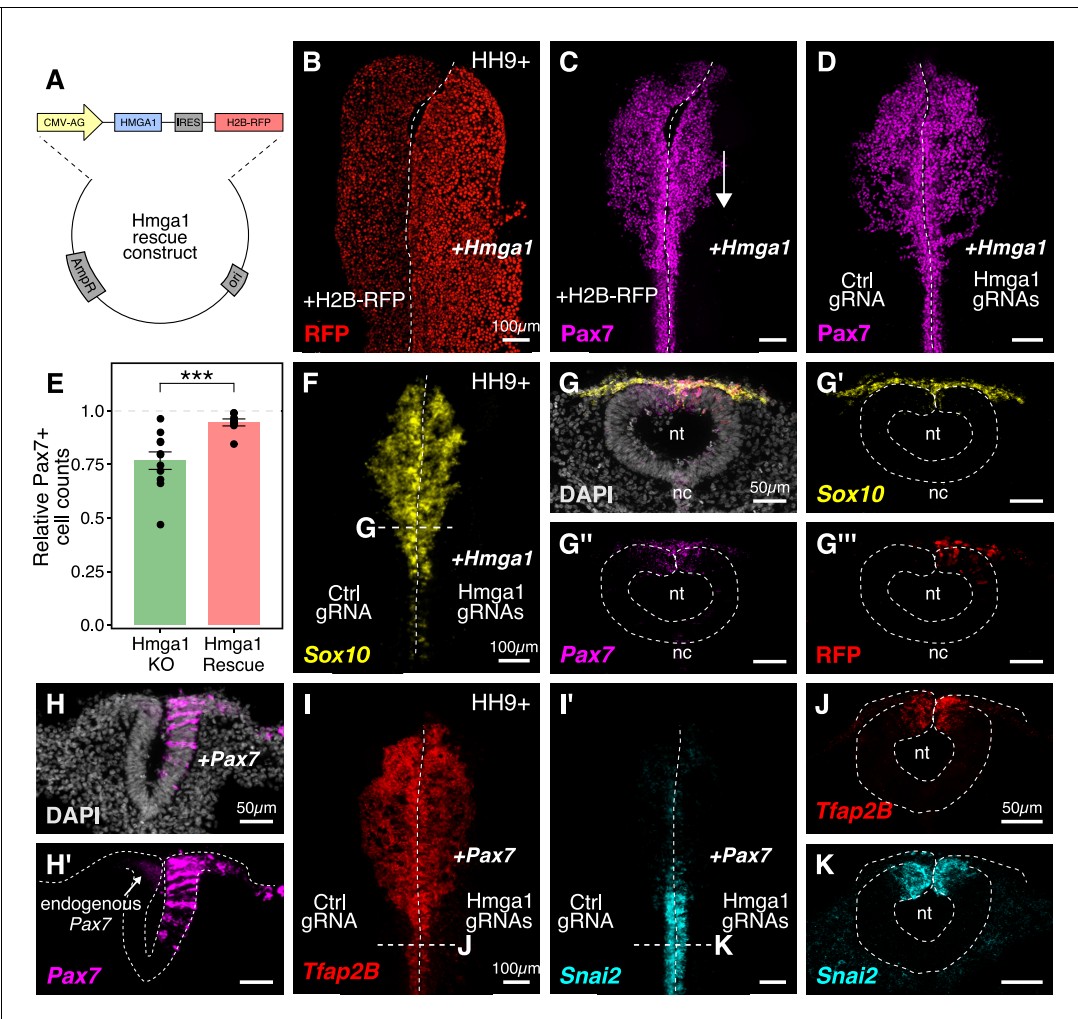

**Figure 5.** Ectopic expression of Hmga1 or Pax7 rescues cranial neural crest specification. (A) Plasmid construct used to rescue *Hmga1*. An independent ribosome entry site (IRES) controls translation of nuclear RFP in transfected cells. The electroporation strategy for knocking out *Hmga1* using CRISPR-Cas9 in gastrula stage chick embryos. (B) Embryos were electroporated with the 'rescue' construct on the right side and a control nuclear RFP plasmid on the left side. (C) Exogenous expression of the *Hmga1* coding sequence under the regulation of a ubiquitous enhancer/promoter combination causes cranial neural crest migration defects. (D–G''') Overexpression of the coding sequence (G''') of *Hmga1* compensates for its loss of function, rescuing proper cranial neural crest migration (D), as assayed by number of Pax7-positive neural crest cells (E), and expression of the neural crest specifier gene *Sox10* (F) in migratory cranial neural crest (G,G') and *Pax7* in the dorsal neural tube (G''). Electroporated embryos were allowed to develop until HH9+ and screened for the expression of H2B-RFP. (H–H') The coding sequence for Pax7 was ectopically expressed in an *Hmga1*-knockout background. Transverse section through a representative embryo shows the comparison between endogenous (left) and overexpressed (right; arrow) *Pax7* transcript levels in the dorsal neural tube. I-K. Ectopic expression of *Pax7* rescued neural crest specification defects caused by the loss of *Hmga1* as assayed by the expression of neural crest specifier genes *Tfap2b* (I) and *Snai2* (I') in transverse cross-sections through the hindbrain (J,K). nt, neural tube; nc, notochord. See also *Figure 5—source data 1*.

The online version of this article includes the following source data for figure 5:

**Source data 1.** Pax7-positive cell counts in transverse sections through the hindbrain compared between Control and *Hmga1*-rescue sides in panel E.

gastrula stage embryos and cultured them ex ovo until HH9+. The left side served as an internal control and was electroporated with an equal concentration of a construct encoding nuclear RFP alone (*Figure 5B*). The results show that, rather than having the opposite effect to loss of function, overexpression of Hmga1 also resulted in a notable reduction in Pax7 expression on the experimental side (*Figure 5C*). This suggests that maintaining appropriate levels of Hmga1 is critical for proper neural crest specification.

The gold standard to demonstrate specificity for loss-of-function experiments is to perform a rescue. We posited that if modulating the levels of Hmga1 was important for neural crest formation, then exogenous expression of Hmga1 in an *Hmga1*-knockout background should successfully rescue neural crest cell numbers. Since the Protospacer Adjacent Motifs (PAMs) adjacent to both *Hmga1* gRNAs are located in the introns (*Figure 3—figure supplement 1A*), the coding sequence on the plasmid would be guarded against the endonuclease activity of Cas9. To test our hypothesis, we knocked out *Hmga1* using CRISPR plasmids as described above, but co-electroporated the 'rescue' construct on the right side. The left side was electroporated with an equal concentration of a plasmid encoding nuclear RFP. Embryos cultured to HH9+ and immunostained for Pax7 revealed that overexpression of Hmga1 concomitant with knocking out the endogenous gene successfully rescued Pax7 levels in cranial neural crest cells (*Figure 5D*). To quantify the extent of rescue, we calculated the ratio of the number of Pax7+ cells on the experimental versus control sides in both wholemount 'knockout' (from *Figure 3G*) and 'rescue' embryos (*Figure 5D*). In unperturbed embryos, this ratio will be close to 1, reflecting similar numbers of Pax7+ cells on both sides of the embryo. However, in the 'knockout' group, we observed a mean ratio of 0.767 ± 0.041 (n = 11), which was significantly different from the ratio of 0.946 ± 0.016 (n = 8) observed in the 'rescue' group (p<0.001, Welch two-sample t-test) (*Figure 5E*, *Figure 5—source data 1*). Next, we probed for the expression of the neural crest specifier gene *Sox10* to ask if rescuing the expression of Hmga1 truly restored the process of neural crest specification. To do this, we processed 'rescue' embryos for HCR against *Sox10* (*Figure 5F*) together with *Pax7* (*Figure 5G*). Indeed, restoring the levels of Hmga1 was sufficient to rescue the expression of *Sox10* (*Figure 5G'*) and *Pax7* (*Figure 5G''*) in early migrating crest, and *Pax7* in the premigratory crest residing in the dorsal neural tube, as visualized in transverse sections through the embryo. We also confirmed that the expression of the 'rescue' construct was restricted to the dorsal neural tube (*Figure 5G'''*), thereby precluding unintended effects on neural tube development.

Finally, given that the loss of *Hmga1* specifically affected Pax7 expression in neural crest precursors, we asked if exogenous expression of Pax7 would be sufficient to rescue the effects of losing *Hmga1* on neural crest specification. We tested this by overexpressing the coding sequence of Pax7 (*Roellig et al., 2017*) on the right side of gastrula stage embryos together with CRISPR plasmids targeting *Hmga1* (*Figure 5H,H'*). Given that the effect of CRISPR-plasmid-mediated loss of *Hmga1* was more penetrant posteriorly, we developed 'Pax7-rescue' embryos to HH9+, processed them for HCR against the neural crest specifier genes *Tfap2b* (*Figure 5I*) and *Snai2* (*Figure 5I'*), and generated transverse sections through the hindbrain. *Tfap2b* is expressed in delaminating and migrating neural crest cells (*Simoes-Costa and Bronner, 2016*), whereas *Snai2* is expressed in premigratory neural crest and is eventually downregulated as the cells begin to migrate (*Taneyhill et al., 2007*). Compared to the *Hmga1*-knockout embryos in which *Tfap2b* mRNA (*Figure 3K*) and Snail2 protein (*Figure 3I'*) levels were notably downregulated, restoring the levels of *Pax7* in an *Hmga1* knockout background rescued the expression of both *Tfap2b* (*Figure 5J*) and *Snai2* (*Figure 5K*) in the dorsal hindbrain. Together, these results suggest that maintaining the correct levels of Hmga1 is necessary for proper neural crest specification in a Pax7-dependent manner.

## Hmga1 activates Wnt signaling to mediate neural crest emigration

Neural crest induction, specification, and emigration from the neural tube are intricate processes that require an interplay between Wnt, FGF, and BMP signaling pathways (*Piacentino and Bronner, 2018*; *Woda et al., 2003*) working reiteratively at different stages of development. For example, at the onset of neural crest emigration, *Wnt1* is prominently expressed in the dorsal neural tube, where premigratory neural crest cells reside (*Simões-Costa et al., 2015*). As a result, these cells turn on *Snai2*, a critical regulator of EMT (*Nieto et al., 1994*) known to function downstream of the Wnt signaling pathway.

After knocking out *Hmga1* using CRISPR plasmids, we noted not only perturbed emigration but also a dramatic reduction in Snail2 levels, even within the subset of Pax7-expressing cells in the dorsal neural folds (*Figure 3I'*). These results raised the intriguing possibility that this might be due to an effect on Wnt signaling in already-specified premigratory neural crest cells. Accordingly, we hypothesized that Hmga1 may function as a Wnt activator in these cells. If so, its loss would be predicted to result in reduced Wnt signaling in the dorsal neural tube. To test this possibility, we used a reporter construct expressing nuclear GFP under the regulation of six Tcf/Lef binding sites and a minimal promoter as a readout for canonical Wnt signaling (*Ferrer-Vaquer et al., 2010*; *Figure 6A*). Plasmids encoding Cas9, gRNAs targeting *Hmga1*, nuclear RFP, and Tcf/Lef:H2B-GFP were electroporated on the right side of gastrula stage embryos, while the left control side was electroporated with plasmids encoding Cas9, control gRNA, nuclear RFP, and Tcf/Lef:H2B-GFP. As described above, this plasmid-based knockout strategy resulted in the loss of *Hmga1* after neural crest specification in the midbrain but well before their emigration. Embryos were allowed to develop until HH9, by which time neural crest cells have started delaminating from the neural tube at the midbrain level (*Figure 6B*). Consistent with our hypothesis, the results show that *Hmga1* knockout caused a notable reduction in canonical Wnt reporter activity on the knockout side compared to the control side (*Figure 6C*) at the midbrain level. Interestingly, these embryos had a neural crest migration defect (*Figure 6—figure supplement 1C–D*) but no notable difference in the number of Pax7+ cells between the two sides (*Figure 6—figure supplement 1E'*), as expected if the *Hmga1* knockout occurred after specification was complete; this is consistent with previous work showing that perturbation of canonical Wnt signaling following specification does not affect the number of Pax7+ cells at cranial EMT stages (*Hutchins and Bronner, 2018*). Quantitation of this effect revealed a significant reduction in reporter activity following the loss of *Hmga1* (*Figure 6D*, *Figure 6—source data 1*; p<0.01; student's t-test), as measured by comparing the ratio between the knockout and the control sides of transfected cells (RFP+) that turned on Wnt signaling within the Pax7+ domain, therefore expressing GFP. While this ratio was expected to be one for embryos with unperturbed Wnt signaling on both sides, we observed a mean ratio of 0.325 ± 0.082 (n = 5), suggesting that Wnt activity was disrupted in the absence of *Hmga1*.

Next, to investigate the mechanism by which *Hmga1* regulates Wnt signaling, we turned to a recently published cranial neural crest chromatin accessibility dataset (*Williams et al., 2019*) and looked for open chromatin regions surrounding genes that encode for known Wnt ligands. In particular, *Wnt1* expression in the dorsal neural tube is known to be necessary for proper delamination of cranial neural crest cells (*Simões-Costa et al., 2015*). Interestingly, we discovered a putative enhancer downstream of *Wnt1* (*Figure 6—figure supplement 1A*) that contained an AT-rich domain consistent with Hmga1-binding motifs (*Figure 6—figure supplement 1B*; *Reeves and Nissen, 1990*). Therefore, we hypothesized that *Hmga1* may modulate Wnt signaling by regulating *Wnt1* expression. To test this, we knocked out *Hmga1* on the right side of gastrula stage embryos using CRISPR plasmids, cultured them ex ovo until HH9, and examined *Wnt1* mRNA expression using in situ hybridization. Indeed, the dorsal neural tube expression of *Wnt1* was severely downregulated (*Figure 6E*) in the midbrain. Consistent with the effect of losing *Hmga1* after neural crest specification, the number of Pax7+ cells in the dorsal neural tube appeared unchanged (*Figure 6E'*). In contrast, no change in *Wnt1* expression was observed at the hindbrain level (*Figure 6F*) which, being developmentally 'younger,' instead exhibited a specification defect that resulted in fewer Pax7+ cells in the dorsal neural tube on the experimental side compared to the control side (*Figure 6F'*). Interestingly, following *Hmga1* knockout, we also observed defects in basement membrane remodeling and laminin channel formation at midbrain levels (*Figure 6G*), another Wnt-dependent process necessary for neural crest EMT; consistent with Wnt inhibition via Draxin overexpression (*Hutchins and Bronner, 2019*), loss of *Hmga1* abrogated laminin remodeling and resulted in physical blockage of the channel through which migrating cranial neural crest cells normally transit (*Figure 6H*). Together, these data indicate that after neural crest specification, *Hmga1* is necessary for the expression of *Wnt1* and activation of canonical Wnt signaling in the dorsal neural tube, and by extension, Wnt-dependent neural crest delamination/EMT.

Finally, given that Hmga1 functions as a canonical Wnt pathway activator, we asked if the migration defects caused by the loss of *Hmga1* post-specification can be rescued by restoring canonical Wnt signaling in premigratory neural crest cells. To address this, we expressed GFP-tagged, stabilized ß-catenin (NC1-Δ90 ß-cat) to upregulate canonical Wnt signaling output specifically in

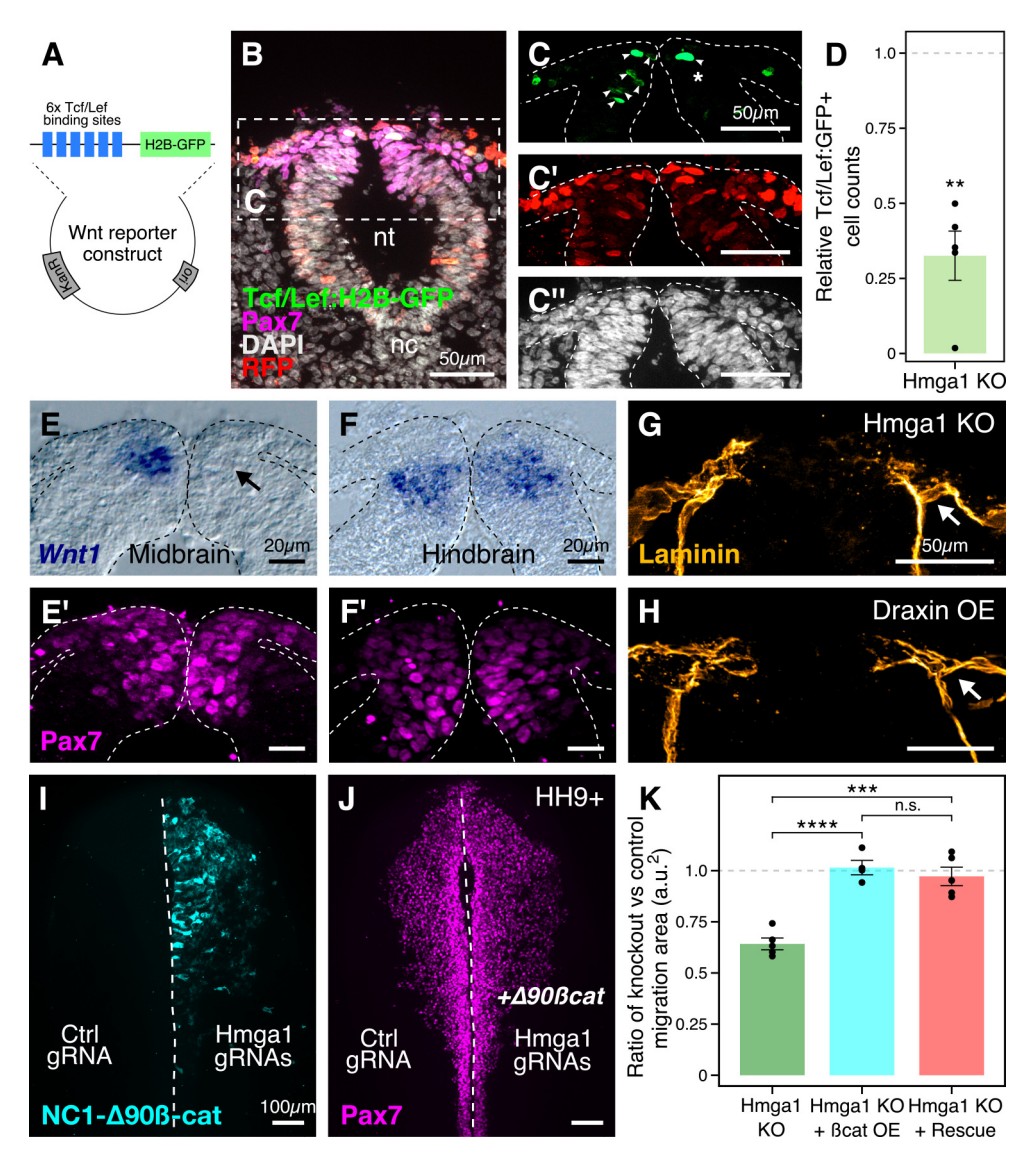

**Figure 6.** Hmga1 activates Wnt signaling pathway in delaminating neural crest cells. (**A**) Plasmid construct used as a readout for Wnt activity (after *Ferrer-Vaquer et al., 2010*). Six TCF/Lef-binding sites together with a minimal promoter regulate the expression of nuclear GFP in transfected cells in response to Wnt signaling. (**B**) Transverse section through the midbrain of a representative HH9+ embryo immunostained for Pax7, GFP, RFP, and DAPI. (**C–C''**) Individual channels of image in B focused on the dorsal neural tube. In the absence of *Hmga1*, Wnt reporter activity was downregulated, resulting in fewer cells that expressed nuclear GFP (arrowheads) on the right side (**C**), even though cells were uniformly transfected on both experimental and control sides (**C'**), and the thickness of the neural tube remained unaffected (**C''**). (**D**) The reduction in Wnt reporter output was quantified as a ratio of number of cells that expressed nuclear GFP, and the number of cells that were successfully transfected and therefore expressed nuclear RFP. The observed difference in GFP+/RFP+ ratio between the knockout and control sides was statistically significant (**p<0.01, student's t-test). (**E–F**) In situ hybridization against *Wnt1* in an *Hmga1*-knockout background. Transverse section through the midbrain (**E**) and hindbrain (**F**) shows reduced and unchanged levels of *Wnt1* ligand in the dorsal neural tube on the experimental (arrow) versus control neural tubes, respectively. (**E'–F'**) The number of Pax7-positive cells in the midbrain appeared unchanged (**E'**), while a reduction was observed in the hindbrain (**F'**). (**G–H**) Transverse section through a representative embryo where *Hmga1* was knocked out using CRISPR plasmids (**G**), and an embryo where Draxin was ectopically expressed (**H**) on the right side, immunostained for Laminin. Similar to Draxin overexpression, *Hmga1* loss resulted in a failure of basement membrane remodeling due to reduced canonical Wnt signaling in neural crest cells, causing the laminin channel to remain blocked on the experimental side (arrows). (**I–J**) Expression of stabilized ß-catenin (NC1-Δ90ß-cat) in

*Figure 6 continued on next page*

*Figure 6 continued*

delaminating cranial neural crest of *Hmga1*-knockout embryos was sufficient to rescue the migration defect. (**K**) Quantification of area covered by cranial neural crest cells on the experimental versus control sides. In the absence of *Hmga1*, cranial neural crest cells fail to migrate properly, a defect that can be separately rescued in *Hmga1*-knockout background by overexpression (OE) of stabilized ß-catenin in delaminating cranial neural crest, or exogenous expression of Hmga1 coding sequence ectopically (Rescue). nt, neural tube; nc, notochord; KO, knockout; OE, overexpression. See also *Figure 6—figure supplement 1*, *Figure 6—source data 1*, *Figure 6—source data 2*.

The online version of this article includes the following source data and figure supplement(s) for figure 6:

**Source data 1.** Ratio of Tcf/Lef:GFP-positive and RFP-positive cell counts in transverse sections through the midbrain compared between Control and *Hmga1*-knockout sides in panel D.

**Source data 2.** Ratio of neural crest cell migration area between experimental and control sides in whole mount embryos compared between *Hmga1*-knockout, ß-catenin-overexpression, and *Hmga1*-rescue conditions in panel K.

**Figure supplement 1.** Hmga1 regulates *Wnt1* expression in premigratory cranial neural crest cells.

premigratory neural crest cells, thus circumventing the critical process of neural crest induction at the neural plate border (*Hutchins and Bronner, 2018*). If loss of *Hmga1* in premigratory neural crest cells resulted in migration defects due to reduced canonical Wnt signaling, expression of a stabilized ß-catenin would be predicted to restore those levels, thereby rescuing proper migration. To test this, we knocked out *Hmga1* on the right side of gastrula stage embryos using CRISPR plasmids as previously described, but co-electroporated NC1-Δ90ß-cat-GFP on the right side. The left side was electroporated with control reagents (*Figure 6I*). Embryos were cultured ex ovo until HH9+ and processed for immunohistochemistry against Pax7. Consistent with our hypothesis that Hmga1 functions as a Wnt activator in neural crest cells, expression of stabilized ß-catenin was sufficient to rescue proper cranial neural crest migration from *Hmga1* knockdown (*Figure 6J*). We also calculated the ratio of the area occupied by migrating cranial neural crest cells between the experimental and control sides (*Figure 6K*, *Figure 6—source data 2*). For wildtype embryos, this ratio would be close to 1, reflecting equal neural crest migration on both sides of the embryo. However, after the loss of *Hmga1*, the ratio of areas on experimental versus control sides was $0.642 \pm 0.028$ (n = 5). Importantly, co-expression of stabilized ß-catenin in premigratory neural crest rescued migration, with an average migration ratio of $1.015 \pm 0.035$ (n = 4), which was significantly different from the knockout group ($p < 0.0001$, ANOVA and post hoc Tukey HSD). Similarly, ubiquitous expression of the Hmga1 coding sequence also rescued neural crest migration, with an average migration ratio of $0.972 \pm 0.044$ (n = 5), which was also significantly different from the knockout group ($p < 0.001$, ANOVA and *post hoc* Tukey HSD). Taken together, our results suggest that Hmga1 mediates the process of EMT by activating canonical Wnt signaling in premigratory neural crest cells, thus enabling them to emigrate from the neural tube.

## Discussion

While chromatin modifiers are known to influence gene expression and cell fate decisions at many stages of development (*Cai et al., 2014*; *Laugesen and Helin, 2014*; *Miller and Hendrich, 2018*; *O'Shaughnessy-Kirwan et al., 2015*), parsing cell-type-specific functions and targets for these proteins is often challenging due to broad expression across multiple tissues and time points. In this study, we have used scRNA-seq to identify a chromatin-remodeling protein, Hmga1, as highly expressed in neural crest cells. Using high-resolution *in situ* HCR and temporally controlled knockdowns, we present evidence for a dual role of Hmga1 in the formation and migration of neural crest cells. At early stages, we find that the neural plate border gene *Pax7* is a downstream target of Hmga1, such that loss of *Hmga1* blocks neural crest specification in a manner that can be rescued by restoring Pax7 expression. After neural crest specification is complete in the closing neural tube, Hmga1 plays a second role in modulating canonical Wnt signaling via alterations in the levels of *Wnt1* in premigratory neural crest cells. This in turn influences neural crest EMT and delamination from the dorsal neural tube (*Figure 7*).

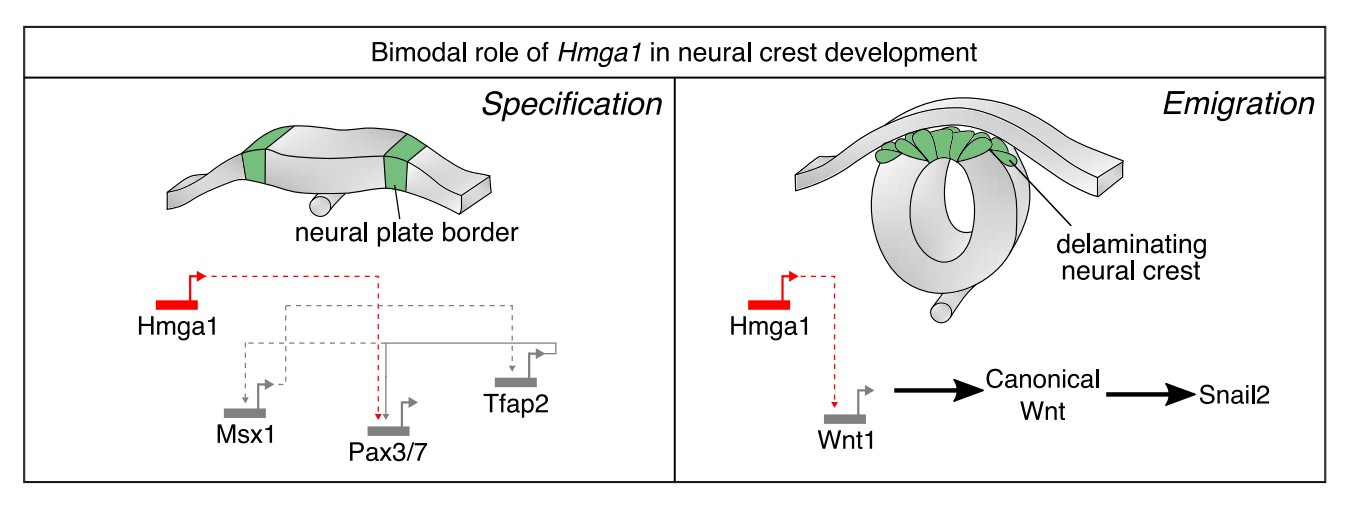

**Figure 7.** Current model for *Hmga1* function in chick neural crest development. Our data suggest that *Hmga1* plays temporally distinct roles in the neural plate border and dorsal neural tube. At the neural plate border, *Hmga1* acts upstream of *Pax7* and is required for proper neural crest specification. Later, in the dorsal neural tube, *Hmga1* regulates the levels of *Wnt1*, thereby modulating the levels of canonical Wnt signaling to control neural crest delamination and subsequent migration.

The canonical Wnt signaling pathway is a major input in a complex GRN that activates transcriptional circuits and controls neural crest specification and cell lineage decisions (*Martik and Bronner, 2017*; *Simões-Costa and Bronner, 2015*; *Williams et al., 2019*), influencing multiple aspects of neural crest development from induction at the neural plate border to proliferation, onset of migration via EMT, and differentiation (*Milet and Monsoro-Burq, 2012*; *Rabadán et al., 2016*; *Simões-Costa and Bronner, 2015*; *Steventon et al., 2009*; *Wu et al., 2003*; *Yanfeng et al., 2003*). For example, regulation of the levels of canonical Wnt signaling is critical for progressive basement membrane remodeling during neural tube closure and neural crest delamination. Consequently, perturbation of Wnt signaling output at different stages of basement membrane remodeling or delamination causes severe defects in neural crest EMT (*Hutchins and Bronner, 2019*; *Rabadán et al., 2016*). Interestingly, early inhibition of canonical Wnt signaling in gastrula stage chick embryos has been shown to reduce *Pax7* expression (*Basch et al., 2006*), whereas canonical Wnt inhibition after neural crest specification does not alter Pax7 expression but has a marked effect on EMT (*Hutchins and Bronner, 2018*). This suggests that there are separable early versus late effects of canonical Wnt signaling during neural crest development.

Hmga1 has been shown to upregulate canonical Wnt signaling components and downstream targets in the intestinal stem cell niche, thereby amplifying signaling output (*Xian et al., 2017*) presumably through increased promotor accessibility. Our results are consistent with a similar role for *Hmga1* in the neural crest, where its loss resulted in decreased output from a canonical Wnt reporter (*Figure 6*) as well as downregulation of the *Wnt1* ligand (*Figure 6*) and the canonical Wnt target Snail2 (*Figure 3*). However, temporally controlled knockdowns revealed that loss of *Hmga1* reduced *Wnt1* expression following completion of neural crest specification, but not at earlier stages of neural crest induction. Conversely, *Hmga1* knockdown affected *Pax7* expression during neural crest induction but not after specification is complete. One possible explanation is that once open/remodeled, the chromatin landscape surrounding the *Pax7* regulatory regions form topologically associating domains (TADs) that are stable and resistant to repression. Alternatively, other Hmga1-independent *cis*-elements may influence *Pax7* expression following neural crest induction. Given that neural crest cells appear to have highly dynamic chromatin accessibility surrounding spatiotemporally regulated enhancer elements (*Williams et al., 2019*), we would predict the latter, although further investigation is needed to distinguish between these possibilities.

An intriguing possibility is that Hmga1, in addition to regulating neural crest EMT, is necessary for maintaining the broad developmental potential of neural crest cells. Neural crest cells exhibit stem-cell-like properties, including multipotency and self-renewal. Thus, the early expression of *Hmga1* in neural crest precursors, together with its reported role in maintaining stemness and self-renewal properties in various stem cell and cancer systems (*Battista et al., 2003*; *Schuldenfrei et al., 2011*; *Shah and Resar, 2012*), may indicate a role in maintaining neural crest stemness. In the dorsal neural tube, self-renewal within the premigratory neural crest is driven by the transcription factor c-Myc (*Kerosuo and Bronner, 2016*), which together with its binding partner regulates cell cycle progression. Consistent with this possibility, we found that the mRNA expression of *c-Myc* is downregulated in the dorsal neural tube following Hmga1 knockout. However, given that canonical Wnt signaling also regulates stem cell self-renewal (*Xu et al., 2016*), as well as cell cycle progression in neural crest (*Burstyn-Cohen et al., 2004*), Hmga1 may have additional Wnt-dependent and/or Wnt-independent roles in the maintenance of the neural crest stem cell pool.

Hmga1 is a member of the high motility group A (HMGA) family of genes that are characterized by their A/T-hook domains and the ability to transform chromatin architecture to regulate transcription of target genes. To date, two members of the HMGA family, *Hmga1* and *Hmga2*, have been identified in mammals (*Reeves et al., 2001*), each having distinct roles in oncogenesis (*Jun et al., 2015*; *Meyer et al., 2007*; *Miyazawa et al., 2004*). As both genes share several common targets, they may compensate for each other where they overlap. Indeed, *Hmga1*/*Hmga2* double knockout mice have severe embryonic lethality, compared to less-penetrant effects in individual knockouts (*Federico et al., 2014*). While both *Hmga1* and *Hmga2* genes are annotated in the chick genome, our scRNA-seq results show that a significant proportion of neural crest cells express *Hmga1* but not *Hmga2*. Consistent with this, loss of *Hmga1* resulted in neural crest-related developmental defects, making it unlikely that there is redundancy and/or compensation by *Hmga2* in chick embryos. Interestingly, only the *hmga2* paralogue is present in *Xenopus laevis* embryos (*Macrì et al., 2016*), but morpholino-mediated knockdown of *hmga2* did not affect the expression of neural plate border gene *pax3/7*. This contrasts with our results in chick embryos, where the loss of *Hmga1* affected both Pax7 transcription and protein levels in the neural plate border. Together with the absence of Hmga2+ cells in our single-cell data, this raises the possibility that individual HMGA family members play discrete roles in neural crest development, similar to their distinct roles in tumorigenesis.

In addition to Hmga1, other chromatin remodeling proteins serve similar functions in neural crest cell fate decisions. For example, both the ATP-dependent chromatin remodeler CHD7 and the histone demethylase Jumonji D2A (KDM4A/JmjD2A) are necessary for expression of neural crest specifier genes; notably, however, these chromatin modifiers appear to function at later developmental time points than Hmga1, as neither knockdown of KDM4A in chick embryos nor CHD7 in *Xenopus* embryos affect *Pax3/7* expression levels (*Bajpai et al., 2010*; *Strobl-Mazzulla et al., 2010*). Furthermore, KDM4A influences *Snai2* and *Wnt1* levels, raising the possibility that Hmga1 may act in concert with KDM4A or other chromatin remodelers to restructure the accessibility of neural crest GRN circuits at different cell fate checkpoints.

In summary, our data reveal a dual role for the chromatin remodeler Hmga1 during neural crest development. First, during specification at the neural plate border, Hmga1 regulates the completion of neural crest induction as assayed by readout of *Pax7* expression at the neural plate border. Later, following neural crest specification, Hmga1 plays a second role in modulating the levels of the canonical Wnt signaling pathway in the closing dorsal neural tube to influence neural crest EMT and delamination at the onset of their migration. Post-embryonically, neural-crest-derived cells are prone to metastasis and give rise to numerous cancers (*Maguire et al., 2015*). Furthermore, neural crest and cancer cells often employ similar mechanisms to drive EMT; in particular, the hallmarks of metastasis often involve disruption of the basement membrane which may also be driven by canonical Wnt signaling (*Gallik et al., 2017*; *Powell et al., 2013*). Interestingly, high expression levels of Hmga1 have been associated with premature EMT and prolonged stemness in several cancers of the pancreas (*Abe et al., 2000*), breast (*Flohr et al., 2003*), lung (*Sarhadi et al., 2006*), and ovaries (*Masciullo et al., 2003*). Therefore, it is interesting to note that Hmga1 may play parallel roles in neural crest development and cancer metastasis. Understanding how Hmga1, and chromatin remodeling in general, alters cell fate decisions and EMT through signaling and transcriptional regulation in neural crest cells will undoubtedly have important and broad implications in human development and disease.

# Materials and methods

## Key resources table

| Reagent type (species) or resource | Designation | Source or reference | Identifiers | Additional information |
|---|---|---|---|---|
| Gene (*Gallus gallus*) | *Hmga1* | UCSC genome browser | NM_204369.1 | |
| Strain, strain background (*Gallus gallus*) | *G. gallus* | Sun State Ranch (Monrovia, CA, USA) | | |
| Antibody | Mouse IgG1 anti-Pax7 | Developmental Studies Hybridoma Bank | RRID:AB_528428 | 1:10 |
| Antibody | Rabbit anti-Laminin | Sigma-Aldrich | RRID:AB_477163 | 1:1000 on sections |
| Antibody | Mouse IgM anti-HNK1 | Developmental Studies Hybridoma Bank | RRID:AB_2314644 | 1:5 |
| Antibody | Rabbit anti-RFP | MBL | RRID:AB_591279 | 1:500 |
| Antibody | Rabbit anti-Slug (C19G7) | Cell Signaling | RRID:AB_2239535 | 1:200 |
| Antibody | Goat IgG anti-GFP | Rockland | RRID:AB_218182 | 1:500 |
| Antibody | Rabbit anti-cleaved-caspase-3 | R and D systems | RRID:AB_2243952 | 1:500 on sections |
| Antibody | Mouse anti-phospho-histone H3 | Abcam | RRID:AB_443110 | 1:500 on sections |
| Recombinant DNA reagent | pCI-H2B-RFP (plasmid) | *Betancur et al., 2010* | | |
| Recombinant DNA reagent | CAG > nls-Cas9-nls (plasmid) | *Gandhi et al., 2017* | RRID:Addgene_99138 | |
| Recombinant DNA reagent | cU6.3>Ctrl. gRNA.f+e (plasmid) | *Gandhi et al., 2017* | RRID:Addgene_99140 | |
| Recombinant DNA reagent | cU6.3>Hmga1.1. gRNA.f+e (plasmid) | This paper | | Detailed in Materials and methods section 'CRISPR-Cas9-mediated perturbations' |
| Recombinant DNA reagent | cU6.3>Hmga1.2. gRNA.f+e (plasmid) | This paper | | Detailed in Materials and methods section 'CRISPR-Cas9-mediated perturbations' |
| Recombinant DNA reagent | FoxD3-NC2:eGFP (plasmid) | *Simões-Costa et al., 2012* | | |
| Recombinant DNA reagent | Tcf/Lef: H2B-GFP (plasmid) | *Ferrer-Vaquer et al., 2010* | RRID:Addgene_32610 | |
| Recombinant DNA reagent | NC1-Δ90β-cat (plasmid) | *Hutchins and Bronner, 2018* | | |
| Recombinant DNA reagent | pCI-Pax7-IRES-H2B-RFP (plasmid) | *Roellig et al., 2017* | | |
| Sequence-based reagent | Hmga1.1. gRNA | This paper | PCR primer | 5'-gCAGGAAGAAACCGGAGgta |
| Sequence-based reagent | Hmga1.2. gRNA | This paper | PCR primer | 5'-GCCAGCTCCAAAGGCAGGgt |
| Sequence-based reagent | Ascl-V5-Fwd | This paper | PCR primer | 5'-ggcgcgccacc ATGGCTGGTAAGCCTA |
| Sequence-based reagent | V5-Hmga1-Fwd | This paper | PCR primer | 5'-CTCCTCGGTCTCGATTCT agcgacgccggcgccaagcc |
| Sequence-based reagent | Hmga1OLP-V5-Rev | This paper | PCR primer | 5'-ggcttggcgccggcgtcgct AGAATCGAGACCGAGGAG |
| Sequence-based reagent | Hmga1-ClaI-Rev | This paper | PCR primer | 5'-ttatcgattcactgctcctcctcggatg |

*Continued on next page*

*Continued*

| Reagent type (species) or resource | Designation | Source or reference | Identifiers | Additional information |
|---|---|---|---|---|
| Sequence-based reagent | Hmga1.1 short guide oligo | This paper | PCR primer | 5'-GCGTAATACGACTCACTATAGG CAGGAAGAAACCGGAGGTAGTTT TAGAGCTAGAAATAGC |
| Sequence-based reagent | Hmga1.2 short guide oligo | This paper | PCR primer | 5'-GCGTAATACGACTCACTATAG GCCAGCTCCAAAGGCAGGGTGT TTTAGAGCTAGAAATAGC |
| Sequence-based reagent | Control short guide oligo | *Hutchins and Bronner, 2018* | PCR primer | Detailed in Materials and methods section 'CRISPR-Cas9-mediated perturbations' |
| Sequence-based reagent | gRNA Primer 1 | *Hutchins and Bronner, 2018* | PCR primer | Detailed in Materials and methods section 'CRISPR-Cas9-mediated perturbations' |
| Sequence-based reagent | gRNA Primer 2 | *Hutchins and Bronner, 2018* | PCR primer | Detailed in Materials and methods section 'CRISPR-Cas9-mediated perturbations' |
| Sequence-based reagent | Guide-constant oligo | *Hutchins and Bronner, 2018* | PCR primer | Detailed in Materials and methods section 'CRISPR-Cas9-mediated perturbations' |
| Commercial assay or kit | Chromium Single Cell 3' Library and Gel Bead Kit v2 | 10X Genomics | Cat# PN-120267 | |
| Commercial assay or kit | Chromium Single Cell A Chip Kit | 10X Genomics | Cat# PN-1000009 | |
| Commercial assay or kit | Endofree maxi prep kit | Macharey Nagel | Cat# 740426.50 | |
| Commercial assay or kit | Agencourt AMPure XP beads | Beckman Coulter | Cat# A63880 | |
| Commercial assay or kit | Dynabeads MyOne SILANE | 10X Genomics | Cat# 2000048 | |
| Commercial assay or kit | SPRIselect Reagent Kit | Beckman Coulter | Cat# B23318 | |
| Commercial assay or kit | High Sensitivity DNA Kit | Agilent | Cat# 5067–4626 | |
| Commercial assay or kit | Qubit dsDNA HS Assay Kit | Thermo Fisher Scientific | Cat# Q32854 | |
| Software, algorithm | Fiji | *Schindelin et al., 2012* | RRID:SCR_002285 | https://imagej.net/Fiji |
| Software, algorithm | Seurat | *Butler et al., 2018* | RRID:SCR_007322 | https://satijalab.org/seurat/ |
| Software, algorithm | Inkscape | Inkscape | RRID:SCR_014479 | https://inkscape.org/ |
| Software, algorithm | Cellranger | 10X Genomics | | |
| Software, algorithm | 2100 Expert software | Agilent | RRID:SCR_014466 | |
| Other | Fluoromount-G | Southern Biotech | Cat# 0100–01 | |
| Other | DAPI | Thermo Fisher Scientific | Cat# D1306 | 1:10000 on sections |

## Electroporations

Chicken embryos (*Gallus gallus*) were commercially obtained from Sun Valley farms (CA), and developed to the specified Hamburger-Hamilton (HH) (*Hamburger and Hamilton, 1951*) stage in a

humidified 37°C incubator. For ex ovo electroporations, embryos were dissected from eggs at HH4, injected with specified reagents, then electroporated as described previously (*Sauka-Spengler and Barembaum, 2008*). Following electroporation, embryos were cultured in fresh albumin/1% penicillin-streptomycin at 37°C and grown to specified HH stages. Once the embryos reached the desired stages, they were screened for transfection efficiency and overall health. Unhealthy and/or poorly transfected embryos were discarded and not included for downstream assays.

## Single-cell suspension

HH4 embryos electroporated with FoxD3-NC2:eGFP were cultured until HH12 ex ovo at 37°C, following which the hindbrain region spanning rhombomeres 6, 7, and 8 was dissected under a fluorescence microscope. For dissociation, several different conditions were tested (dissociation in a glass dish for 1 hr in Accumax (EMD Millipore), chemical dissociation on a nutator for 15, 30, and 45 min, and chemical dissociation with gentle pipetting for 15, 30, and 45 min). The quality of the single-cell suspension obtained was tested by a trypan blue-based live-dead staining. Accordingly, we pooled dissected tissue washed in chilled 1X DPBS and incubated it in Accumax cell dissociation solution for 15 min at 37°C with gentle mixing every 5 min. Dissociation was terminated using Hanks Buffered Saline Solution (HBSS) (Corning) supplemented with BSA Fraction V (Sigma; 0.2% w/v). The suspension was centrifuged at 300 g for 4 min to collect cells at the bottom, the supernatant was removed, and the pellet was resuspended in 1 mL HBSS-BSA. To remove cell debris and clumps, the 1 mL suspension was passed through a 20 µm filter in a clean hood. This suspension was loaded on a 10X Chromium chip A (v2) to generate GEMs. The library was prepared according to the manufacturer's protocol and sequenced on the Illumina HiSeq platform using the paired end chemistry.

## scRNA-seq data analysis

The raw fastq files were aligned to the galgal6 (GRCg6a) genome assembly obtained from the ENSEMBL database using the cellranger pipeline downloaded from the 10X Genomics website. For feature counts, a custom galgal6 GTF file, where all annotated 3' UTRs were extended by 1 kb, was used. This was done to compensate for improper gene annotations in the chick genome. The count matrices were then imported in R for analysis using Seurat (*Butler et al., 2018*). The initial filtering step discarded all cells with fewer than 200 and more than 10,000 genes per cell. We also filtered out cells expressing more than 5% mitochondrial or less than 20% ribosomal genes. Next, we removed genes corresponding to small RNAs, micro RNAs, mitochondria, and general housekeeping from the count matrix. Following log normalization and Principal Component Analysis, the cells were clustered using the first 15 dimensions (calculated from the elbow plot). Different resolution parameter values were tested, and a value of 0.45 was used to identify subpopulations within the data. Dimensional reductionality was performed using the UMAP (*McInnes et al., 2018*) algorithm. All plots were created in R, exported in SVG format, and assembled in Inkscape.

## Hybridization chain reaction

HCR v3 was performed using the protocol suggested by Molecular Technologies (*Choi et al., 2018*) with minor modifications. Briefly, the embryos were fixed in 4% paraformaldehyde (PFA) overnight at 4°C or 2 hr at room temperature, washed in 0.1% PBS-Tween, dehydrated in a series of 25%, 50%, 75%, and 100% methanol washes, and incubated overnight at −20°C in 100% methanol. The next day, the embryos were rehydrated, treated with proteinase-K for 2–2.5 min, and incubated with 10 pmol of probes dissolved in hybridization buffer overnight at 37°C. The next day, following several washes in 'probe wash buffer,' embryos were incubated in 30 pmol of hairpins H1 and H2 diluted in Amplification buffer at room temperature overnight. The next morning, embryos were washed in 0.1% 5x-SSC-Tween and imaged. All probes were designed and ordered through Molecular Technologies.

## Molecular cloning

The coding sequence of Hmga1 was obtained from the UCSC genome browser (*Karolchik et al., 2003*). A V5 tag was cloned in-frame at the N-terminus using overlap PCR (Accuprime). The fusion product was cloned downstream of the CAGG promoter, upstream of the IRES-H2B-RFP segment of pCI-H2B-RFP (*Betancur et al., 2010*) to clone the final plasmid (CAGG >V5-HMGA1-IRES-H2B-RFP).

The Cas9 and gRNA constructs (*Gandhi et al., 2017*), neural crest enhancer FoxD3-NC2:eGFP (*Simões-Costa et al., 2012*), canonical Wnt reporter TCF/Lef:H2B-GFP (*Ferrer-Vaquer et al., 2010*), and neural crest-specific stabilized ß-catenin NC1-Δ90ß-cat (*Hutchins and Bronner, 2018*) have all been previously described and validated.

## CRISPR-Cas9-mediated perturbations

The genomic locus for *Hmga1* was obtained from the UCSC genome browser (*Karolchik et al., 2003*). Two gRNAs targeting the coding sequence, the first targeting exon 3 (5'-CCAGGAA-GAAACCGGAGgta-3'), and the second targeting exon 4 (5'-GCCAGCTCCAAAGGCAGGgt-3'), were designed using CHOPCHOP (*Labun et al., 2019*). The protospacers were cloned downstream of the chick U6.3 promoter as described in *Gandhi et al., 2017*. For control electroporations, the control gRNA described in *Gandhi et al., 2017* was used. *CAGG >nls-Cas9-nls* and *CAGG >H2* B-RFP were electroporated at a concentration of 2 µg/µl, together with either 0.75 µg/µl per *Hmga1* gRNA on the right side or 1.5 µg/µl of control gRNA on the left side.

For Cas9/in vitro-transcribed gRNA RNP experiments, we generated single-guide RNAs (sgRNAs) as described previously (*Hutchins and Bronner, 2018*), using the following primers:

> Hmga1.1 short guide oligo:
> 5'-GCGTAATACGACTCACTATAGGCAGGAAGAAACCGGAGGTAGTTTTAGAGCTAGAAA
> TAGC-3';
> Hmga1.2 short guide oligo: 5'-
> GCGTAATACGACTCACTATAGGCCAGCTCCAAAGGCAGGGTGTTTTAGAGCTAGAAATAGC
> ;
> Control short guide oligo:
> 5'-GCGTAATACGACTCACTATAGGCACTGCTACGATCTACACCGTTTTAGAGCTAGAAA
> TAGC;
> gRNA Primer 1: 5'-CACGCGTAATACGACTCACTATAG;
> gRNA Primer 2: 5'-AAAGCACCGACTCGGTGCCAC;
> Guide-constant oligo: 5'-
> AAAGCACCGACTCGGTGCCACTTTTTCAAGTTGATAACGGACTAGCCTTATTTTAACTTGC
> TATTTCTAGCTCTAAAAC.

Of the recombinant Cas9 (M0646; New England Biolabs), 2.6 µl was mixed with equal volumes of control gRNA or 1.3 µl each of the two *Hmga1* gRNAs, and heated to 37°C for 15 min. The solution was then incubated at room temperature for 15 min, mixed with 2 µg/µl H2B-RFP and 1 µl of steril-ized 2% food dye, and injected in embryos for electroporation.

## In situ hybridization and immunohistochemistry

Chromogenic in situ hybridization was performed as described previously for *FoxD3, Sox10,* and *Tfap2b, c-Myc,* and *Wnt1* (*Kerosuo and Bronner, 2016*; *Simoes-Costa and Bronner, 2016*; *Simões-Costa et al., 2015*).

Immunohistochemistry was performed as described previously (*Gandhi et al., 2017*). Briefly, embryos were fixed for 20 min at room temperature, blocked in 10% goat or donkey serum in 0.5% PBS-Triton overnight at 4°C, incubated overnight at 4°C in primary antibodies diluted in blocking solution, washed at room temperature in 0.5% PBS-Triton, incubated overnight at 4°C in secondary antibodies diluted in blocking solution, washed at room temperature in 0.5% PBS-Triton, and proc-essed for imaging and/or cryosectioning. The following primary antibodies and concentrations were used: Mouse IgM HNK1 (1:5; Developmental Studies Hybridoma Bank (3H5)); Mouse IgG1 Pax7 (1:10; Developmental Studies Hybridoma Bank (RRID:AB_528428)); Goat GFP (1:500; Rockland Cat# 600-101-215); Rabbit RFP (1:500; MBL Cat# PM005); Rabbit Snail2 (1:200; Cell Signaling Technology (9585)); Rabbit Laminin (1:1000; Sigma-Aldrich (L9393)); Rabbit cleaved-Caspase 3 (1:500; R and D Systems Cat# AF835); Mouse phospho-histone H3 (1:500; Abcam Cat# Ab14955). The following spe-cies-specific secondary antibodies labeled with Alexa Fluor dyes (Invitrogen) were used: Goat/Don-key anti-Mouse Alexa Fluor 647 (for Pax7 and pH3; 1:250), Goat/Donkey Goat anti-Mouse IgM Alexa Fluor 350/488 (for HNK1; 1:250), Goat/Donkey anti-Rabbit Alexa Fluor 488 (for Snail2, cleaved-Cas-pase3, and Laminin; 1:250), Donkey anti-goat Alexa Fluor 488 (for Citrine; 1:500), and Goat/Donkey anti-rabbit Alexa Fluor 568 (for RFP; 1:500).

## Cryosectioning

Following whole mount imaging, embryos were washed in 5% and 15% sucrose overnight at 4°C. The next day, embryos were transferred to molten gelatin for 3–5 hr at 37°C, embedded in molds at room temperature, frozen in liquid nitrogen, and stored at −80°C overnight. Embedded embryos were sectioned on a micron cryostat to obtain 16 µm sections through immunostained embryos and 20 µm sections through in situ hybridized embryos. The sections were degelatinized at 42°C in 1x PBS for 5 min, washed in 1x PBS, soaked in 1x PBS containing 0.1 µg/mL DAPI for 2 min, washed in 1x PBS and distilled water. Fluoromount mounting medium was used to mount coverslips on slides.

## Microscope image acquisition, analysis, and statistical tests

Whole mount embryos and sections on slides were imaged on a Zeiss Imager M2 with an ApoTome module and/or Zeiss LSM 880 confocal microscope at the Caltech Biological Imaging Facility. Images were post-processed using FIJI imaging software (*Schindelin et al., 2012*). To calculate corrected total cell fluorescence (CTCF), the following formula was used:

$$CTCF = Integrated\ Density - (Selected\ area * Mean\ background\ fluorescence)$$

For cell counts, a median filter was applied to 8-bit images. A Bernsen-based auto local-thresholding method (*Bernsen, 1986*) followed by watershed segmentation was used to identify cell boundaries. The 'Analyze particles' function was used to count the number of cells. All statistical analyses were performed in R. The Wilcoxon rank test was used in datasets that were not normally distributed. In cases where the underlying distribution was normal, a student's t-test was used to calculate significance. In cases where multiple samples were compared, Analysis of Variance (ANOVA) test combined with Tukey HSD correction was used. Post hoc power analysis was used to validate sample size and confirm sufficient statistical power (>0.8).

## Acknowledgements

For technical assistance, we thank Fan Gao with the Caltech Bioinformatics Resource Center of the Beckman Institute, Giada Spigolon and Andres Collazo with the Caltech Biological Imaging facility of the Beckman Institute, and Sisi Chen and Paul Rivaud with the Single Cell Profiling and Engineering Center (SPEC) of the Beckman Institute. We thank members of the Bronner lab for helpful discussions.

## Additional information

### Competing interests

Marianne E Bronner: Senior editor, *eLife*. The other authors declare that no competing interests exist.

### Funding

| Funder | Grant reference number | Author |
| --- | --- | --- |
| National Institutes of Health | R01DE027568 | Marianne E Bronner |
| National Institutes of Health | R01HL14058 | Marianne E Bronner |
| National Institutes of Health | R01DE027538 | Marianne E Bronner |
| American Heart Association | 18PRE34050063 | Shashank Gandhi |
| National Institutes of Health | K99DE028592 | Erica J Hutchins |

The funders had no role in study design, data collection and interpretation, or the decision to submit the work for publication.

## Author contributions
Shashank Gandhi, Conceptualization, Resources, Software, Formal analysis, Supervision, Validation, Investigation, Visualization, Methodology, Writing - original draft, Writing - review and editing; Erica J Hutchins, Resources, Validation, Investigation, Visualization, Writing - review and editing; Krystyna Maruszko, Investigation, Visualization, Writing - original draft; Jong H Park, Matthew Thomson, Resources, Methodology; Marianne E Bronner, Conceptualization, Supervision, Funding acquisition, Writing - review and editing

## Author ORCIDs
Shashank Gandhi (ID) https://orcid.org/0000-0002-4081-4338
Erica J Hutchins (ID) http://orcid.org/0000-0002-4316-0333
Marianne E Bronner (ID) https://orcid.org/0000-0003-4274-1862

## Decision letter and Author response
Decision letter https://doi.org/10.7554/eLife.57779.sa1
Author response https://doi.org/10.7554/eLife.57779.sa2

## Additional files

### Supplementary files
• Transparent reporting form

### Data availability
Sequencing data files have been deposited on NCBI under the accession number PRJNA624258.

The following dataset was generated:

| Author(s) | Year | Dataset title | Dataset URL | Database and Identifier |
|---|---|---|---|---|
| Gandhi S, Hutchins EJ, Maruszko K, Park JH, Thomson M, Bronner ME | 2020 | Single cell RNA sequencing of the chick hindbrain | http://www.ncbi.nlm.nih.gov/bioproject/?term=PRJNA624258 | NCBI BioProject, PRJNA624258 |

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
