## [Decision Letter]

Thank you for submitting your article "Bimodal function of chromatin remodeler *Hmga1* in neural crest induction and Wnt-dependent emigration" for consideration by *eLife*. Your article has been reviewed by three peer reviewers, including Lukas Sommer as the Reviewing Editor and Reviewer #1, and the evaluation has been overseen by Didier Stainier as the Senior Editor. The following individuals involved in review of your submission have agreed to reveal their identity: Filippo M Rijli (Reviewer #2); Igor Adameyko (Reviewer #3).

The reviewers have discussed the reviews with one another and the Reviewing Editor has drafted this decision to help you prepare a revised submission.

Summary:

In this interesting study, Gandhi et al. used single cell transcriptomics on early migrating neural crest cells to show that the non-histone chromatin remodeler *Hmga1* is highly expressed in neural crest cells. *Hmga1* has been overlooked in previous bulk sequencing approaches due to its simultaneous expression in other cell types. Using a series of elegant CRISPR-Cas9-based temporally controlled knockdowns (KDs), high resolution in situ hybridization, and rescue experiments in the chick embryo, they convincingly show that *Hmga1* has essential and temporally distinct roles in neural crest development.

This study introduces a new major regulator of neural crest development and represents another example in the literature of a developmental program being re-expressed in carcinogenesis. This is a very interesting, comprehensive and well-done study.

Specific points to be experimentally addressed:

1) Part of the phenotype (including the reduction of *Pax7* domain) can be caused by the increased apoptosis of KO cells. This needs to be checked. Specifically, cell death at the neural plate border should be evaluated for *Hmga1* KD vs. control in embryos electroporated on both sides.

2) HMGA1 is known as a powerful driver of cell cycle progression. The KO phenotype can be partly explained by potentially reduced proliferation. This should be investigated more in depth with EdU labeling and / or other complementary methods.

Specific points to be addressed either experimentally (optionally) or by text revision:

1) The early role of *Hmga1* involves *Pax7* regulation already in the neural plate border (Figure 4). Are other neural plate border genes (as previously defined by the Bronner lab) affected as well? If so, do they possibly share putative *Hmga1* binding motifs in support of a broader role of *Hmga1* in neural crest specification? Or is the phenotype mainly mediated by *Hmga1*-dependent control of *Pax7* and can be rescued by *Pax7* overexpression?

2) Gain of function experiments with overexpression of HMGA1/2 might be beneficial.

3) Can the authors speculate on how *Hmga1* may function as a Wnt activator? This point should be discussed.

4) The role of HMGA1 in tissue growth and proliferation should be mentioned in the Introduction section.

5) In general, the figure legends should include more details. Specifically, abbreviations should be explained such as for example in Figure 2. Moreover, the structures/cells, arrows and arrowheads point to, should be consistently referred to in the figure legends.

6) In the third paragraph of the subsection “*Hmga1* regulates expression of *Pax7* in neural crest precursors in the neural plate border”, it is stated that the loss of *Hmga1* reduces both endogenous *Pax7* protein expression and reporter expression, however the representative images in Figure 4J/J' do not clearly show this effect. Rather, the number of cells appear to be reduced.

7) In the third paragraph of the subsection “*Hmga1* activates Wnt signaling to mediate neural crest emigration”, it is mentioned that *Hmga1* knockdown causes defects in basement membrane remodelling and channel formation. However, this is not very obvious from the pictures shown in Supplementary Figure 4D-E. Maybe the authors want to include a positive control here, such as overexpression of draxin that was shown to prevent channel formation, as shown in Hutchins and Bronner, 2019.

8) Wnt pathway activation was rescued by expression of a stabilised β-catenin (subsection “*Hmga1* activates Wnt signaling to mediate neural crest emigration”, fourth paragraph), and this resulted in proper neural crest migration in *Hmga1* knockouts. The effect on neural crest migration is not very clear in Figure 5F. An image of the area covered by migratory neural crest cells should be included here, as similarly done in Figure 6—figure supplement 1D.

Revisions expected in follow-up work:

1) It would be important to see if the neural crest cells with HMGA1 KO are capable of differentiating into traditional derivatives or not. What happens to these cells at later developmental stages?

2) Since the authors take advantage of a single cell transcriptomics, it might be beneficial to attempt to sequence a KO condition to measure how the reduction of HMGA2 correlates with the cell cycle phases, induction of delamination and migratory phenotype. This is a large experiment, which is not strictly necessary for this revision (given the situation with COVID 19). We would like to mention this approach in case the authors will have the opportunity and resources to tackle this now or later in the follow up projects.

---

## [Author Response]

Revisions for this paper:Specific points to be experimentally addressed:1) Part of the phenotype (including the reduction of Pax7 domain) can be caused by the increased apoptosis of KO cells. This needs to be checked. Specifically, cell death at the neural plate border should be evaluated for Hmga1 KD vs. control in embryos electroporated on both sides.

The reviewers raise a valid possibility. To test if apoptosis might have occurred after loss of *Hmga1*, we electroporated Cas9 protein-*Hmga1* gRNA RNP complexes on the right side of gastrula stage embryos and Cas9-control gRNA RNP complexes on the left side, and immunostained transverse sections through the neural plate border for cleavedCaspase-3 as a readout for cell death. No significant difference was noted between the treated and control sides. These data have been added to the revised manuscript [Figure 4—figure supplement 1, subsection “*Hmga1* regulates expression of *Pax7* in neural crest precursors in the neural plate border”, last paragraph].

2) HMGA1 is known as a powerful driver of cell cycle progression. The KO phenotype can be partly explained by potentially reduced proliferation. This should be investigated more in depth with EdU labeling and / or other complementary methods.

We agree this is a valid concern. To test possible effects on the cell cycle, we immunostained sections through the neural plate border of embryos after loss of *Hmga1* on one side (as described above) with an antibody against phospho-Histone H3 to serve as a readout for cell proliferation. Quantification of these data revealed no significant difference between the treated and control sides, suggesting that reduced proliferation does not explain the observed loss of *Pax7* in the neural plate border cells. These data have been added to the revised manuscript [Figure 4—figure supplement 1, subsection “*Hmga1* regulates expression of *Pax7* in neural crest precursors in the neural plate border”, last paragraph].

Specific points to be addressed either experimentally (optionally) or by text revision:1) The early role of Hmga1 involves Pax7 regulation already in the neural plate border (Figure 4). Are other neural plate border genes (as previously defined by the Bronner lab) affected as well? If so, do they possibly share putative Hmga1 binding motifs in support of a broader role of Hmga1 in neural crest specification? Or is the phenotype mainly mediated by Hmga1-dependent control of Pax7 and can be rescued by Pax7 overexpression?

We thank the reviewers for this insightful comment. We performed two experiments to test this, first by looking at the effects of losing *Hmga1* on other neural plate border markers, and second, by asking whether *Pax7* could rescue the effects of losing *Hmga1* on neural crest specification.

To check if the loss of *Hmga1* affected other neural plate border genes, we knocked out *Hmga1* using Cas9 protein-*Hmga1* gRNA RNP complexes on the right side of gastrula stage embryos and processed embryos for Hybridization Chain Reaction (HCR) against the neural plate border genes *Msx1* and *Tfap2a*. The results show that while *Pax7* transcripts were reduced on the knockout side, the expression of *Msx1* and *Tfap2a* was unaffected. This suggests that *Hmga1* specifically regulates *Pax7* in neural crest precursors. These data are now included in the revised manuscript [Figure 4K-Q, subsection “*Hmga1* is not required for expression of neural plate border genes *Msx1* or *Tfap2a*”].

Next, we tested whether the effects of *Hmga1* loss could be rescued by *Pax7*. To this end, we knocked out *Hmga1* on the right side of gastrula stage embryos using CRISPR plasmids, and simultaneously expressed *Pax7* using a previously described plasmid construct (Roellig et al., 2017) on the right side. The embryos were processed for HCR against neural crest specifier genes *Tfap2B* and *Snai2*. Indeed, expressing *Pax7* was sufficient to rescue the expression of these two genes in the dorsal neural tube as observed in cross-sections through the hindbrain. We thank the reviewers for this suggestion; these data have been added to the revised manuscript [Figure 5H-K, subsection “*Hmga1* and *Pax7* rescue the effects of losing *Hmga1* on neural crest specification”, last paragraph].

2) Gain of function experiments with overexpression of HMGA1/2 might be beneficial.

We thank the reviewers for this suggestion. To address this question, we overexpressed *Hmga1* on the right side of gastrula stage embryos and processed embryos for antibody staining against *Pax7*. Interestingly, overexpression of *Hmga1* results in a cranial neural crest migration defects. We reasoned that this was a result of precocious Wnt signaling activation (see below), which in turn results in maintenance of Cadherin6B at premigratory stages, causing a delamination defect. This phenotype has been previously documented by our lab in embryos where the Wnt antagonist Draxin was knocked down using morpholinos and/or CRISPR-Cas9 (Hutchins and Bronner, 2018, 2019). These data are now included in the revised manuscript [Figure 5, subsection “*Hmga1* and *Pax7* rescue the effects of losing *Hmga1* on neural crest specification”].

3) Can the authors speculate on how Hmga1 may function as a Wnt activator? This point should be discussed.

This is a critical question, and we thank the reviewers for raising it. In our first submission, we speculated that *Hmga1* may regulate the levels of *Wnt1* transcription. This was based on finding an AT-rich domain, to which *Hmga1* might bind, in a putative *Wnt1* cranial neural crest enhancer isolated from a recently published study (Williams et al., 2019). To test this hypothesis, we performed additional experiments in which we knocked out *Hmga1* on the right side of gastrula stage embryos using CRISPR plasmids, allowed embryos to develop to HH9, and processed them for in situ hybridization against *Wnt1*. Interestingly, the results show that loss of *Hmga1* reduces *Wnt1* transcripts in the midbrain, further elaborating a possible mechanism by which *Hmga1* activates Wnt signaling in emigrating cranial neural crest cells in a manner independent of neural crest specification. In these embryos, while the number of *Pax7*+ cells remained similar, more cells appeared stuck in the dorsal neural tube on the knockout side, consistent with a Wnt-mediated migration defect. These data have been added to the revised manuscript [Figure 6, Figure 6—figure supplement 1, subsection “*Hmga1* activates Wnt signaling to mediate neural crest emigration”], and we have expanded on this discussion point [Discussion].

4) The role of HMGA1 in tissue growth and proliferation should be mentioned in the Introduction section.

We thank the reviewers for this suggestion. We now discuss the role *Hmga1* plays in tissue growth in early mouse embryos [subsection “*Hmga1* is expressed in the neural plate, neural plate border, and neural crest cells”, first paragraph].

5) In general, the figure legends should include more details. Specifically, abbreviations should be explained such as for example in Figure 2. Moreover, the structures/cells, arrows and arrowheads point to, should be consistently referred to in the figure legends.

We thank the reviewers for pointing out this oversight. The figure legends have now been expanded to include more details about the figure panels. All arrows and arrowheads have now been described clearly for each figure.

6) In the third paragraph of the subsection “Hmga1 regulates expression of Pax7 in neural crest precursors in the neural plate border”, it is stated that the loss of Hmga1 reduces both endogenous Pax7 protein expression and reporter expression, however the representative images in Figure 4J/J' do not clearly show this effect. Rather, the number of cells appear to be reduced.

We thank the reviewers for this comment. Given that the enhancer data was purely used as another way of confirming the loss of *Pax7* and was not important for the conclusions of our study, we decided to remove them from the manuscript in response to the reviewers’ concern.

7) In the third paragraph of the subsection “Hmga1 activates Wnt signaling to mediate neural crest emigration”, it is mentioned that Hmga1 knockdown causes defects in basement membrane remodelling and channel formation. However, this is not very obvious from the pictures shown in Supplementary Figure 4D-E. Maybe the authors want to include a positive control here, such as overexpression of draxin that was shown to prevent channel formation, as shown in Hutchins and Bronner, 2019.

We thank the reviewers for pointing out this shortcoming. To address this, we have now included a representative image showing the effect of Draxin overexpression on basement membrane remodeling [Figure 6, subsection “*Hmga1* activates Wnt signaling to mediate neural crest emigration”].

8) Wnt pathway activation was rescued by expression of a stabilised β-catenin (subsection “Hmga1 activates Wnt signaling to mediate neural crest emigration”, fourth paragraph), and this resulted in proper neural crest migration in Hmga1 knockouts. The effect on neural crest migration is not very clear in Figure 5F. An image of the area covered by migratory neural crest cells should be included here, as similarly done in Figure 6—figure supplement 1D.

We thank the reviewers for this suggestion. As suggested, we now include quantifications of these data by calculating the ratio of area of neural crest migration between treated and control sides. To further strengthen the effect of expressing ß-catenin on neural crest migration, we have compared this ratio between three different experiments (ß-catenin rescue, *Hmga1* knockout, and *Hmga1* rescue) and use ANOVA followed by Tukey HSD to demonstrate that these differences are statistically significant. These results are now included in the revised manuscript [Figure 6K, subsection “*Hmga1* activates Wnt signaling to mediate neural crest emigration”, last paragraph].

Revisions expected in follow-up work:1) It would be important to see if the neural crest cells with HMGA1 KO are capable of differentiating into traditional derivatives or not. What happens to these cells at later developmental stages?

This is an excellent suggestion, and in fact, the first author was hoping to recruit a summer student to perform these experiments, but due to COVID-19, no undergraduates are on campus this summer and the lab is working in shifts. These experiments are in the pipeline and will be addressed in the future when we are back to full operation.

2) Since the authors take advantage of a single cell transcriptomics, it might be beneficial to attempt to sequence a KO condition to measure how the reduction of HMGA2 correlates with the cell cycle phases, induction of delamination and migratory phenotype. This is a large experiment, which is not strictly necessary for this revision (given the situation with COVID 19). We would like to mention this approach in case the authors will have the opportunity and resources to tackle this now or later in the follow up projects.

Once again, this is another excellent suggestion, one that we are hoping to pursue in the near future. The first author is particularly excited about the application of single-cell transcriptomics to address such questions!